# Atmospheric Temperature and CO₂: Hen-Or-Egg Causality?

**Demetris Koutsoyiannis** [1,*] **and Zbigniew W. Kundzewicz** [2]

[1] Department of Water Resources and Environmental Engineering, School of Civil Engineering, National Technical University of Athens, 157 80 Athens, Greece

[2] Institute for Agricultural and Forest Environment, Polish Academy of Sciences, 60-809 Poznań, Poland; kundzewicz@yahoo.com

\* Correspondence: dk@itia.ntua.gr

**Abstract:** It is common knowledge that increasing $CO_2$ concentration plays a major role in enhancement of the greenhouse effect and contributes to global warming. The purpose of this study is to complement the conventional and established theory, that increased $CO_2$ concentration due to human emissions causes an increase in temperature, by considering the reverse causality. Since increased temperature causes an increase in $CO_2$ concentration, the relationship of atmospheric $CO_2$ and temperature may qualify as belonging to the category of "hen-or-egg" problems, where it is not always clear which of two interrelated events is the cause and which the effect. We examine the relationship of global temperature and atmospheric carbon dioxide concentration in monthly time steps, covering the time interval 1980–2019 during which reliable instrumental measurements are available. While both causality directions exist, the results of our study support the hypothesis that the dominant direction is $T \rightarrow CO_2$. Changes in $CO_2$ follow changes in $T$ by about six months on a monthly scale, or about one year on an annual scale. We attempt to interpret this mechanism by involving biochemical reactions as at higher temperatures, soil respiration and, hence, $CO_2$ emissions, are increasing.

**Keywords:** temperature; global warming; greenhouse gases; atmospheric $CO_2$ concentration

---

*Πότερον ἡ ὄρνις πρότερον ἢ τὸ ᾠὸν ἐγένετο* (Which of the two came first, the hen or the egg?).

Πλούταρχος, Ηθικά, Συμποσιακὰ Β, Πρόβλημα Γ (Plutarch, Moralia, Quaestiones convivales, B, Question III).

## 1. Introduction

The phrase "hen-or-egg" is a metaphor describing situations where it is not clear which of two interrelated events or processes is the cause and which the effect. Plutarch was the first to pose this type of causality as a philosophical problem using the example of the hen and the egg, as indicated in the motto above. We note that in the original Greek text, "ἡ ὄρνις" is feminine (article and noun), meaning the hen rather than the chicken. Therefore, here, we preferred the form "hen-or-egg" over "chicken-or-egg", which is more common in English (Very often, in online Greek texts, e.g., [1,2], "ἡ ὄρνις" appears as "ἡ ἄρνις". After extended searching, we contend that this must be an error, either an old one in copying of manuscripts, e.g., by monks in monasteries, or a modern one, e.g., in OCR. We are confident that the correct word is "ὄρνις").

The objective of the paper is to demonstrate that the relationship of atmospheric $CO_2$ and temperature may qualify as belonging to the category of "hen-or-egg" problems. First, we discuss the relationship between temperature and $CO_2$ concentration by revisiting intriguing results from

proxy data-based palaeoclimatic studies, where the change in temperature leads and the change in $CO_2$ concentration follows. Next, we discuss the databases of modern (instrumental) measurements related to global temperature and atmospheric $CO_2$ concentration and introduce a methodology to analyse them. We develop a stochastic framework, introducing useful notions of time irreversibility and system causality while we discuss the logical and technical complications in identifying causality, which prompt us to seek just necessary, rather than sufficient, causality conditions. In the Results section, we examine the relationship of these two quantities using the modern data, available at the monthly time step. We juxtapose time series of global temperature and atmospheric $CO_2$ concentration from several sources, covering the common time interval 1980–2019. In our methodology, it is the timing rather than the magnitude of changes that is important, being the determinant of causality. While logical, physically based arguments support the "hen-or-egg" hypothesis, indicating that both causality directions exist, interpretation of cross-correlations of time series of global temperature and atmospheric $CO_2$ suggests that the dominant direction is $T \rightarrow CO_2$, i.e., the change in temperature leads and the change in $CO_2$ concentration follows. We attempt to interpret this latter mechanism by noting the positive feedback loop—higher temperatures increase soil respiration and, hence, $CO_2$ emissions.

The analysis reported in this paper was prompted by observation of an unexpected (and unfortunate) real-world experiment: during the COVID-19 lockdown in 2020, despite the unprecedented decrease in carbon emissions (Figure 1), there was an increase in atmospheric $CO_2$ concentration, which followed a pattern similar to previous years (Figure 2). Indeed, according to the International Energy Agency (IEA) [3], global $CO_2$ emissions were over 5% lower in the first quarter of 2020 than in that of 2019, mainly due to an 8% decline in emissions from coal, 4.5% from oil, and 2.3% from natural gas. According to other estimates [4], the decrease is even higher: the daily global $CO_2$ emissions decreased by 17% by early April 2020 compared with the mean 2019 levels, while for the whole 2020, a decrease between 4% and 7% is predicted. Despite that, as seen in Figure 2, the normal pattern of atmospheric $CO_2$ concentration (increase until May and decrease in June and July) did not change. Similar was the behaviour after the 2008–2009 financial crisis, but the most recent situation is more characteristic because the COVID-19 decline in 2020 is the most severe ever, even when considering the periods corresponding to World Wars. It is also noteworthy that, as shown in Figure 1, there are three consecutive years in the 2010s where there are no major increases, in emissions while there was an increase in $CO_2$ concentration. (At first glance, this does not sound reasonable and we have therefore cross-checked the data with other sources, namely the Global Carbon Atlas [5], and the database of Our World In Data [6]; we found only slight differences.) Interestingly, Figure 1 also shows a rapid growth in emissions after the 2008–2009 global financial crisis, which is in agreement with Peters et al. [7].

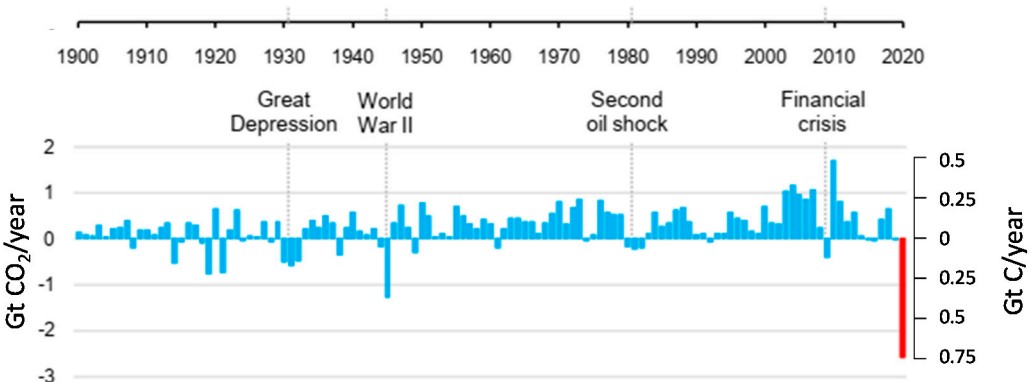

**Figure 1.** Annual change in global energy-related $CO_2$ emissions (adapted from International Energy Agency (IEA) [3]).

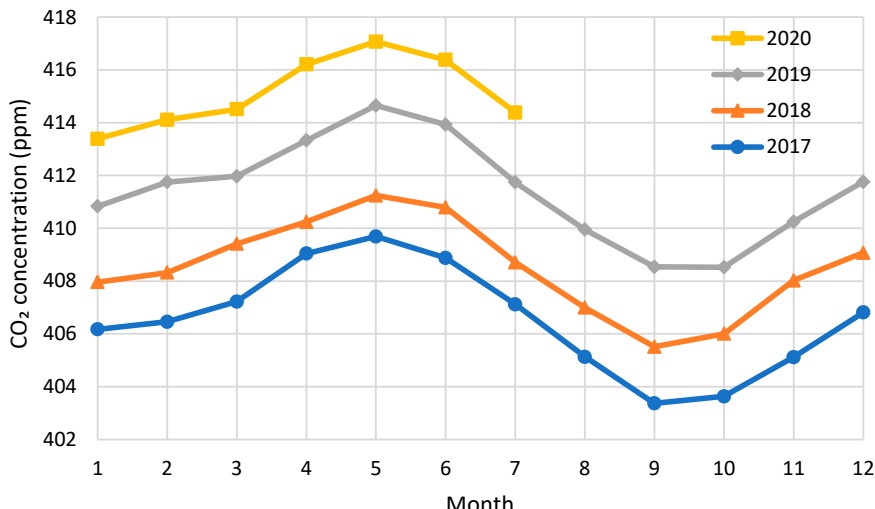

**Figure 2.** Atmospheric $CO_2$ concentration measured in Mauna Loa, Hawaii, USA, in the last four years.

## 2. Temperature and Carbon Dioxide—From Arrhenius and Palaeo-Proxies to Instrumental Data

Does the relationship of atmospheric carbon dioxide ($CO_2$) and temperature classify as a "hen-or-egg"-type causality? If we look at the first steps of studying the link between the two, the reply is clearly negative. Arrhenius (1896, [8]), the scientist most renowned for studying the causal relationship between two quantities, regarded the changes in atmospheric carbon dioxide concentration as the cause and the changes in temperature as the effect. Specifically, he stated:

> *Conversations with my friend and colleague Professor Högbom together with the discussions above referred to, led me to make a preliminary estimate of the probable effect of a variation of the atmospheric carbonic acid* [meaning $CO_2$] *on the temperature of the earth. As this estimation led to the belief that one might in this way probably find an explanation for temperature variations of 5–10 °C, I worked out the calculation more in detail and lay it now before the public and the critics.*

Furthermore, following the Italian meteorologist De Marchi (1895, [9]), whom he cited, he rejected what we call today *Milanković cycles* as possible causes of the glacial periods. In addition, he substantially overestimated the role of $CO_2$ in the greenhouse effect of the Earth's atmosphere. He calculated the relative weights of absorption of $CO_2$ and water vapour as 1.5 and 0.88, respectively, or a ratio of 1:0.6.

Arrhenius [8] also stated that "if the quantity of carbonic acid increases in geometric progression, the augmentation of the temperature will increase nearly in arithmetic progression". This Arrhenius's "rule" (which is still in use today) is mathematically expressed as

$$T - T_0 = \alpha \ln\left(\frac{[CO_2]}{[CO_2]_0}\right) \tag{1}$$

where $T$ and $[CO_2]$ denote temperature and $CO_2$ concentration, respectively, $T_0$ and $[CO_2]_0$ represent reference states, and $\alpha$ is a constant.

Here, it is useful to note that Arrhenius's studies were not the first on the subject. Arrhenius [8] cites several other authors, among whom Tyndall (1865, [10]) for pointing out the enormous importance of atmospheric absorption of radiation and for having the opinion that water vapour has the greatest influence on this. Interestingly, it appears that the first experiments on the ability of water vapour and carbon dioxide to absorb heat were undertaken even earlier by Eunice Newton Foote (1856, [11]), even though she did not recognize the underlying mechanisms or even the distinction of short- and long-wave radiation [12–14]).

While the fact that the two variables are tightly connected is beyond doubt, the direction of the simple causal relationship needs to be studied further. Today, additional knowledge has

been accumulated, particularly from palaeoclimatic studies, which allow us to examine Arrhenius's hypotheses on a sounder basis. In brief, we can state the following:

- Indeed, $CO_2$ plays a substantial role as a greenhouse gas. However, modern estimates of the contribution of $CO_2$ to the greenhouse effect differ largely from Arrhenius's results, attributing 19% of the long-wave radiation absorption to $CO_2$ against 75% of water vapour and clouds (Schmidt et al. [15]), or a ratio of 1:4.
- During the Phanerozoic Eon, Earth's temperature varied by even more than 5–10 °C, which was postulated by Arrhenius—see Figure 3. Even though the link of temperature and $CO_2$ is beyond doubt, this is not clear in Figure 3, where it is seen that the $CO_2$ concentration has varied by about two orders of magnitude and does not always synchronize with the temperature variation. Other factors may become more important at such huge time scales. Thus, an alternative hypothesis of the galactic cosmic ray flux as a climate driver via solar wind modulation has been suggested [16,17], which has triggered discussion or dispute [14,18–23]. The $T$–$CO_2$ relationship becomes more legible and rather indisputable in proxy data of the Quaternary (see Figure 4). It has been demonstrated in a persuasive manner by Roe [24] that in the Quaternary, it is the effect of Milanković cycles (variations in eccentricity, axial tilt, and precession of Earth's orbit), rather than of atmospheric $CO_2$ concentration, that explains the glaciation process. Specifically (quoting Roe [24]),

  *variations in atmospheric $CO_2$ appear to lag the rate of change of global ice volume. This implies only a secondary role for $CO_2$—variations in which produce a weaker radiative forcing than the orbitally-induced changes in summertime insolation—in driving changes in global ice volume.*

Despite falsification of some of Arrhenius's hypotheses, his line of thought remained dominant. Yet, there have been some important studies, based on palaeoclimatological reconstructions (mostly the Vostok ice cores [25,26]), which have pointed to the opposite direction of causality, i.e., the change in temperature as the cause and that of $CO_2$ concentration as the effect. Such claims have been based on the fact that temperature change leads and $CO_2$ concentration change follows. In agreement with Roe [24], several papers have found the time lag to be positive, with estimates varying from 50 to 1000 years or more, depending on the time period and the particular study [27–32]. Claims that $CO_2$ concentration leads (i.e., a negative lag) have not generally been made in these studies. At most, a synchrony claim has been sought on the basis that the estimated positive lags are often within the 95% uncertainty range [31], while in one publication [29], it has been asserted that a "short lead of $CO_2$ over temperature cannot be excluded". With respect to the last deglacial warming, Liu et al. [32], using breakpoint lead–lag analysis, again find positive lags and conclude that the $CO_2$ is an internal feedback in Earth's climate system rather than an initial trigger.

Since palaeoclimatic data suggest a direction opposite to that assumed by Arrhenius, Koutsoyiannis [30], using palaeoclimatic data from the Vostok ice cores at a time resolution of 1000 years and a stochastic framework similar to that of the present study (see Section 4.1), concluded that a change in temperature precedes that of $CO_2$ by one time step (1000 years), as illustrated in Figure 4. He also noted that this causality condition holds for a wide range of time lags, up to 26,000 years, and, hence, the time lag is positive and most likely real. He asserted that the problem is obviously more complex than that of exclusive roles of cause and effect, classifying it as a "hen-or-egg" causality problem. Obviously, however, the proxy character of these data and the overly large time step of the time series reduce the reliability and accuracy of the results.

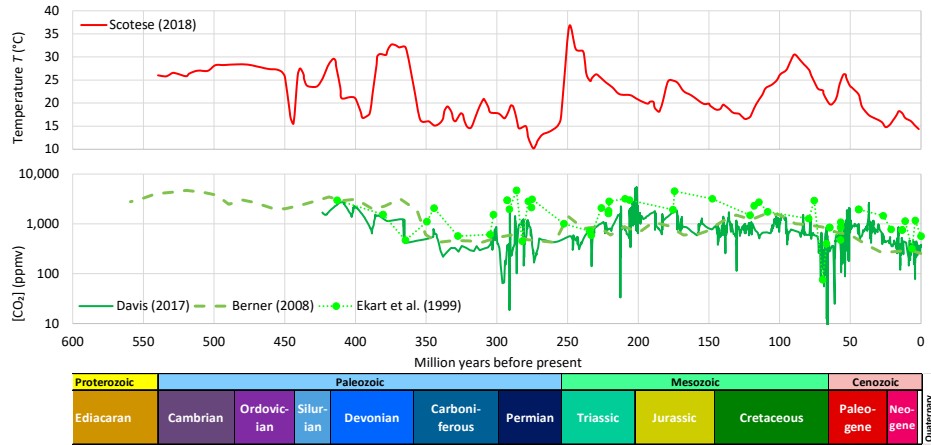

**Figure 3.** Proxy-based reconstructions of global mean temperature and $CO_2$ concentration during the Phanerozoic. The temperature reconstruction by Scotese [33] was mainly based on proxies from [21,34–36], while the $CO_2$ concentration proxies have been taken from Davis [37], Berner [38], and Ekart et al. [39]; all original figures were digitized in this study. The chronologies of geologic eras shown in the bottom of the figure have been taken from the International Commission on Stratigraphy [40].

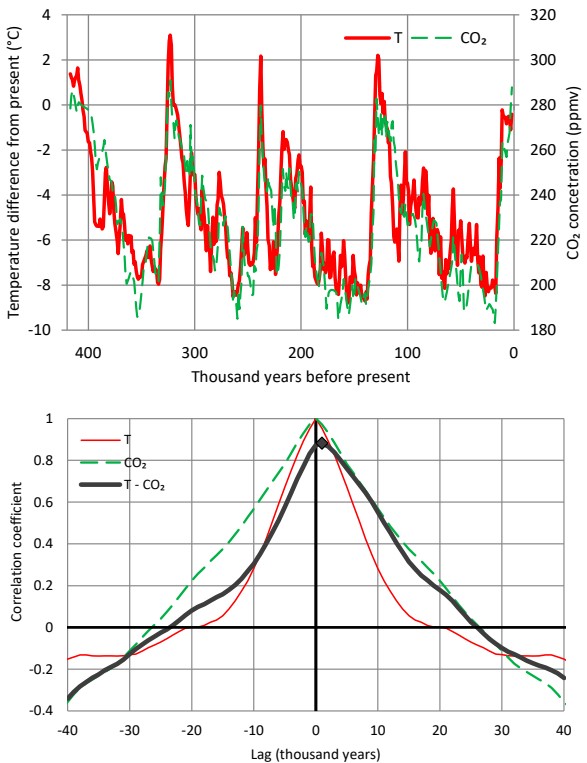

**Figure 4.** (**upper**) Time series of temperature and $CO_2$ concentration from the Vostok ice core, covering part of the Quaternary (420,000 years) with time step of 1000 years. (**lower**) Auto- and cross-correlograms of the two time series. The maximum value of the cross-correlation coefficient, marked as ◆, is 0.88 and appears at lag 1 (thousand years) (adapted from Koutsoyiannis [30]).

Studies exploring the rich body of modern datasets have also been published. Most of the studies have been based on the so-called "Granger causality test" (see Section 4.2). To mention a few, Kodra et al. [41], after testing several combinations and lags within the Granger framework, did not find any statistically significant results at the usual 5% significance level (they only found 2 cases at the

10% significance level; see their Tables 2 and 3). Stern and Kaufmann [42] studied, again within the Granger framework, the causality between radiative forcing and temperature, and found that both natural and anthropogenic forcings cause temperature change, and also that the inverse is true, i.e., temperature causes greenhouse gas concentration changes. They concluded that their results

> *show that properly specified tests of Ganger* [sic] *causality validate the consensus that human activity is partially responsible for the observed rise in global temperature and that this rise in temperature also has an effect on the global carbon cycle.*

By contrast, Stips et al. [43] used a different method [44] to investigate the causal structure and concluded that their

> *study unambiguously shows one-way causality between the total Greenhouse Gases and GMTA* [global mean surface temperature anomalies]. *Specifically, it is confirmed that the former, especially* $CO_2$, *are the main causal drivers of the recent warming.*

Here, we use a different path to study the causal relation between temperature and $CO_2$ concentration with the emphasis given on the exploratory and explanatory aspect of our analyses. While we occasionally use the Granger statistical test, this is not central in our approach. Rather, we place the emphasis on time directionality in the relationship, which we try to identify in the simplest possible manner, i.e., by finding the lag, positive or negative, which maximizes the cross-correlation between the two processes (see Section 4.1). We visualize our results by plots, so as to be simple, transparent, intuitive, readily understandable by the reader, and hopefully persuading. For the algorithmic-friendly reader, we also provide statistical testing results which just confirm what is directly seen in the graphs.

Another difference of our study, from most of the earlier ones, is our focus on changes, rather than current states, in the processes we investigate. This puts the technique of process differencing in central place in our analyses. This technique is quite natural and also powerful for studying time directionality [30]. We note that differencing has also been used in a study by Humlum et al. [45], which has several similarities with our study, even though it is not posed in a formal causality context, as well as in the study by Kodra et al. [41]. However, differencing has been criticized for potentially eliminating long-run effects and, hence, providing information only on short-run effects [42,46]. Even if this speculation were valid, it would not invalidate the differencing technique for the following reasons:

- The short-term effects deserve to be studied, as well as the long-term ones.
- The modern instrumental records are short themselves and only allow the short-term effects to be studied.
- For the long-term effects, the palaeo-proxies provide better indications, as already discussed above.

## 3. Data

Our investigation of the relationship of temperature with concentration of carbon dioxide in the atmosphere is based on two time series of the former process and four of the latter. Specifically, the temperature data are of two origins, satellite and ground-based. The satellite dataset, developed at the University of Alabama in Huntsville (UAH), infers the temperature, $T$, of three broad levels of the atmosphere from satellite measurements of the oxygen radiance in the microwave band using advanced (passive) microwave sounding units on NOAA and NASA satellites [47,48]. The data are publicly available on the monthly scale in the forms of time series of "anomalies" (defined as differences from long-term means) for several parts of earth as well as in maps. Here, we use only the global average on monthly scale for the lowest level, referred to as the lower troposphere. The ground-based data series we use is the CRUTEM.4.6.0.0 global T2m land temperature [49]. This originates from a gridded dataset of historical near-surface air temperature anomalies over land. Data are available for each month from January 1850 to the present. The dataset is a collaborative product of the Met Office

Hadley Centre and the Climatic Research Unit at the University of East Anglia. We note that both sources of information, UAH and CRUTEM, provide time series over the globe, land, and oceans; here, we deliberately use one source for the globe and one for the land.

The two temperature series used in the study are depicted in Figure 5. They are consistent with each other (and correlated, $r = 0.8$), though the CRUTEM4 series shows a larger increasing trend than the UAH series. The differences can be explained by three reasons: (a) the UAH series includes both land and sea, while the chosen CRUTEM4 series is for land only, in which the increasing trend is substantially higher than in sea; (b) the UAH series refers to some high altitude in the troposphere (see details in Koutsoyiannis [50]), while the CRUTEM4 series refers to the ground level; and (c) the ground-based CRUTEM4 series might be affected by urbanization (many ground stations are located in urban areas). In any case, the difference in the increasing trends is irrelevant for the current study, as the timing, rather than the magnitude, of changes is the determinant of causality. This will manifest in our results.

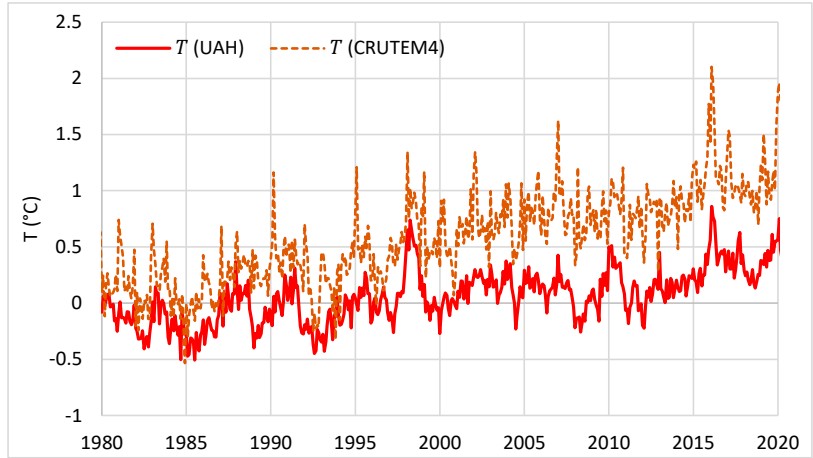

**Figure 5.** Plots of the data series of global temperature "anomalies" since 1980, as used in the study, from satellite measurements over the globe (UAH) and from ground measurements over land (CRUTEM4).

The most famous $CO_2$ dataset is that of Mauna Loa Observatory [51]. The Observatory, located on the north flank of Mauna Loa Volcano on the Big Island of Hawaii, USA, at an elevation of 3397 m above sea level, is a premier atmospheric research facility that has been continuously monitoring and collecting data related to the atmosphere since the 1950s. The NOAA also has other stations that systematically measure atmospheric $CO_2$ concentration, namely at Barrow, Alaska, USA and at South Pole. The NOAA's Global Monitoring Laboratory Carbon Cycle Group also computes global mean surface values of $CO_2$ concentration using measurements of weekly air samples from the Cooperative Global Air Sampling Network. The global estimate is based on measurements from a subset of network sites. Only sites where samples are predominantly of well-mixed marine boundary layer air, representative of a large volume of the atmosphere, are considered (typically at remote marine sea level locations with prevailing onshore winds). Measurements from sites at high altitude (such as Mauna Loa) and from sites close to anthropogenic and natural sources and sinks are excluded from the global estimate. (Details about this dataset are provided in [52]).

The period of data coverage varies, but all series cover the common 40-year period 1980–2019 which, hence, constitute the time reference of all our analyses. As a slight exception, the Barrow (Alaska) and South Pole measurements have not yet been available in final form for 2019 and, thus, this year was not included in our analyses of these two time series. The data of the latter two stations are given in irregular-step time series, which was regularized (by interpolation) to monthly in this study. All other data series have already been available on a monthly scale. While some of the earlier studies

refer to a longer time span (e.g., [42,43] which start from 1850s), here, we avoid using non-systematic data earlier than 1980 due to their low reliability and bypass the raised controversies explained in Appendix A.1.

All four $CO_2$ time series used in the study are depicted in Figure 6. They show a superposition of increasing trends and annual cycles whose amplitudes increase as we head from the South to the North Pole. The South Pole series has opposite phase of oscillation compared to the other three.

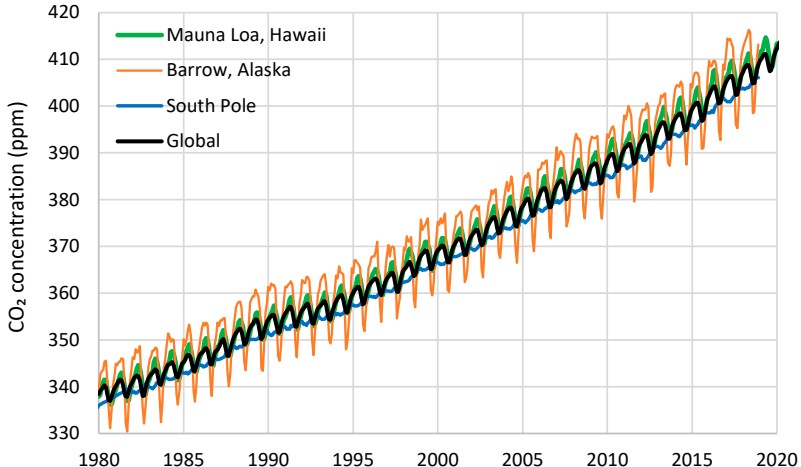

**Figure 6.** Plots of the data series of atmospheric $CO_2$ concentration measured in Mauna Loa (Hawaii, USA), Barrow (Alaska, USA), and South Pole, and the global average.

The annual cycle is better seen in Figure 7, where we have removed the trend with standardization, namely by dividing each monthly value by the geometric average of the preceding 5-year period. The reason why we used division rather than subtraction and geometric rather than arithmetic average (being thus equivalent to subtracting or averaging the logarithms of $CO_2$ concentration) will become evident in Section 5. In the right panel of Figure 7, which depicts monthly statistics of the time series of the left panel, it is seen that in all sites but the South Pole, the annual maximum occurs in May; that of the South Pole occurs in September.

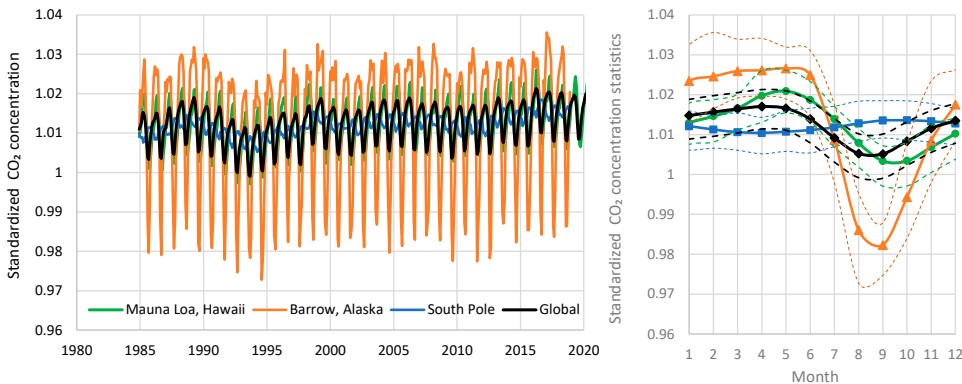

**Figure 7.** Plots of atmospheric $CO_2$ concentration after standardization: (**left**) each monthly value is standardized by dividing with the geometric average of the 5-year period before it. (**right**) Monthly statistics of the values of the left panel; for each month, the average is shown in continuous line and the minimum and maximum in thin dashed lines of the same colour as the average.

## 4. Methods

### 4.1. Stochastic Framework

A recent study [30] investigated time irreversibility in hydrometeorological processes and developed a theoretical framework in stochastic terms. It also studied necessary conditions for causality, which is tightly linked to time irreversibility. A simple definition of time reversibility within stochastics is the following, where underlined symbols denote stochastic (random) variables and non-underlined ones denote values thereof or regular variables.

A stochastic process $\underline{x}(t)$ at continuous time $t$, with $n$th order distribution function:

$$F(x_1, x_2, \ldots, x_n; t_1, t_2, \ldots, t_n) := P\left\{\underline{x}(t_1) \le x_1, \underline{x}(t_2) \le x_2, \ldots, \underline{x}(t_n) \le x_n\right\} \tag{2}$$

is time-symmetric or time-reversible if its joint distribution does not change after reflection of time about the origin, i.e., if for any $n$, $t_1, t_2, \ldots, t_n$,

$$F(x_1, x_2, \ldots, x_n; t_1, t_2, \ldots, t_n) = F(x_1, x_2, \ldots, x_n; -t_1, -t_2, \ldots, -t_n). \tag{3}$$

If times $t_i$ are equidistant, i.e., $t_i - t_{i-1} = D$, the definition can be also written by reflecting the order of points in time, i.e.,

$$F(x_1, x_2, \ldots, x_{n-1}, x_n; t_1, t_2, \ldots, t_{n-1}, t_n) = F(x_1, x_2, \ldots, x_{n-1}, x_n; t_n, t_{n-1}, \ldots, t_2, t_1). \tag{4}$$

A process that is not time-reversible is called time-asymmetric, time-irreversible, or time-directional. Important results related to time (ir)reversibility are the following:

- A time reversible process is also stationary (Lawrance [53]).
- If a scalar process $\underline{x}(t)$ is Gaussian (i.e., all its finite dimensional distributions are multivariate normal) then it is reversible (Weiss [54]). The consequences are (a) a directional process cannot be Gaussian; (b) a discrete-time ARMA process (and a continuous-time Markov process) is reversible if and only if it is Gaussian.
- However, a vector (multivariate) process can be Gaussian and irreversible at the same time. A multivariate Gaussian linear process is reversible if and only if its autocovariance matrices are all symmetric (Tong and Zhang [55]).

Time asymmetry of a process can be studied more conveniently (or even exclusively in a scalar process) through the differenced process, i.e.,

$$\underline{\tilde{x}}_{\tau, \nu} := \underline{x}_{\tau + \nu} - \underline{x}_\tau \tag{5}$$

for an appropriate time-step $\nu$ of differencing. The differenced process represents change of the original process within a time period of length $\nu$. We further define the cumulative process of $\underline{x}_\tau$ for discrete time $\kappa$ as

$$\underline{X}_\kappa := \underline{x}_1 + \underline{x}_2 + \ldots + \underline{x}_\kappa. \tag{6}$$

Through this, we find that the time average of the original process $\underline{x}_\tau$ for discrete time scale $\kappa$ is

$$\underline{x}_\tau^{(\kappa)} := \frac{\underline{x}_{(\tau-1)\kappa+1} + \underline{x}_{(\tau-1)\kappa+2} + \ldots + \underline{x}_{\tau\kappa}}{\kappa} = \frac{\underline{X}_{\tau\kappa} - \underline{X}_{(\tau-1)\kappa}}{\kappa}. \tag{7}$$

Similar equations for the cumulative and averaged processes for the differenced process $\underline{\tilde{x}}_{\tau, \nu}$ are given in Appendix A.2.

The variance of the process $\underline{x}_\tau^{(\kappa)}$ is a function of the time scale $\kappa$, which is termed the climacogram of the process:

$$\gamma_\kappa := \text{var}[\underline{x}_\tau^{(\kappa)}]. \tag{8}$$

The autocovariance function for time lag $\eta$ is derived from the climacogram through the relationship [56]

$$c_\eta = \frac{(\eta+1)^2 \gamma_{|\eta+1|} + (\eta-1)^2 \gamma_{|\eta-1|}}{2} - \eta^2 \gamma_{|\eta|}. \tag{9}$$

For sufficiently large $\kappa$ (theoretically as $\kappa \to \infty$), we may approximate the climacogram as

$$\gamma_\kappa \propto \kappa^{2H-2} \tag{10}$$

where $H$ is termed the *Hurst parameter*. The theoretical validity of such (power-type) behaviour of a process was implied by Kolmogorov (1940 [57]). The quantity $2H$—$2$ is visualized as the slope of the double logarithmic plot of the climacogram for large time scales. In a random process, $H = 1/2$, while in most natural processes, $1/2 \le H \le 1$, as first observed by Hurst (1951 [58]). This natural behaviour is known as (long-term) *persistence* or *Hurst–Kolmogorov* (HK) *dynamics*. A high value of $H$ (approaching 1) indicates enhanced presence of patterns, enhanced change, and enhanced uncertainty (e.g., in future predictions). A low value of $H$ (approaching 0) indicates enhanced fluctuation or *antipersistence* (sometimes misnamed as quasi-periodicity as the period is not constant).

For a stationary stochastic process $\underline{x}_\tau$, the differenced process $\underline{\tilde{x}}_\tau$ has mean zero and variance:

$$\tilde{\gamma}_{\nu,1} := \mathrm{var}[\underline{\tilde{x}}_{\tau,\nu}] = \mathrm{var}[\underline{x}_{\kappa+\nu}] + \mathrm{var}[\underline{x}_\tau] - 2\mathrm{cov}[\underline{x}_{\tau+\nu}, \underline{x}_\tau] = 2(\gamma_1 - c_\nu) \tag{11}$$

where $\gamma_1$ and $c_\nu$ are the variance and lag $\nu$ autocovariance, respectively, of $\underline{x}_\tau$. Furthermore, it has been demonstrated [30] that the Hurst coefficient of the differenced process $\underline{\tilde{x}}_\tau$ precisely equals zero, which means that $\underline{\tilde{x}}_\tau$ is completely antipersistent, irrespective of $\gamma_\kappa$.

To study irreversibility in vector processes, we can use second-order moments and, in particular, cross-covariances among the different components of the vector. In particular (adapting and simplifying the analyses and results in Koutsoyiannis, [30]), given two processes $\underline{x}_\tau$ and $\underline{y}_\tau$, we could study the cross-correlations:

$$r_{\tilde{x}\tilde{y}}[\nu,\eta] := \mathrm{corr}\left[\underline{\tilde{x}}_{\tau,\nu}, \underline{\tilde{y}}_{\tau+\eta,\nu}\right]. \tag{12}$$

Time (ir)reversibility could then be characterized by studying the properties of symmetry or asymmetry of $r_{\tilde{x}\tilde{y}}(\nu,\eta)$ as a function of the time lag $\eta$. In a symmetric bivariate process, $r_{\tilde{x}\tilde{y}}[\nu,\eta] = r_{\tilde{x}\tilde{y}}[\nu,-\eta]$, and if the two components are positively correlated, the maximum of $r_{\tilde{x}\tilde{y}}[\nu,\eta]$ will appear at lag $\eta = 0$. If the bivariate process is irreversible, this maximum will appear at a lag $\eta_1 \ne 0$ and its value will be $r_{\tilde{x}\tilde{y}}[\nu,\eta_1]$.

Time asymmetry is closely related to causality, which presupposes irreversibility. Thus, "no causal process (i.e., such that of two consecutive phases, one is always the cause of the other) can be reversible" (Heller, [59]; see also [60]). In probabilistic definitions of causality, time asymmetry is determinant. Thus, Suppes [61] defines causation thus: "An event $B_{t'}$ [occurring at time $t'$] is a prima facie cause of the event $A_t$ [occurring at time $t$] if and only if (i) $t' < t$, (ii) $P\{B_{t'}\} > 0$, (iii) $P(A_t|B_{t'}) > P(A_t)$". In addition, Granger's [62] first axiom in defining causality reads, "The past and present may cause the future, but the future cannot".

Consequently, in simple causal systems, in which the process component $\underline{x}_\tau$ is the cause of $\underline{y}_\tau$ (like in the clear case of rainfall and runoff, respectively), it is reasonable to expect $r_{\tilde{x}\tilde{y}}[\nu,\eta] \ge 0$ for any $\eta \ge 0$, while $r_{\tilde{x}\tilde{y}}[\nu,\eta] = 0$ for any $\eta = 0$. However, in "hen-or-egg" causal systems, this will not be the case, and we reasonably expect $r_{\tilde{x}\tilde{y}}[\nu,\eta] \ne 0$ for any $\eta$. Yet, we can define a dominant direction of causality based on the time lag $\eta_1$ maximizing cross-correlation. Formally, $\eta_1$ is defined for a specified $\nu$ as

$$\eta_1 := \underset{\eta}{\mathrm{argmax}}\left|r_{\tilde{x}\tilde{y}}(\nu,\eta)\right|. \tag{13}$$

We can thus distinguish the following three cases:

- If $\eta_1 = 0$, then there is no dominant direction.

- If $\eta_1 > 0$, then the dominant direction is $\underline{x}_\tau \rightarrow \underline{y}_\tau$.
- If $\eta_1 < 0$, then the dominant direction is $\underline{y}_\tau \rightarrow \underline{x}_\tau$.

Justification and further explanations of these conditions are provided in Appendix A.3.

### 4.2. Complications in Seeking Causality

It must be stressed that the above conditions are considered as necessary and not sufficient conditions for a causative relationship between the processes $\underline{x}_\tau$ and $\underline{y}_\tau$. Following Koutsoyiannis [30] (where additional necessary conditions are discussed), we avoid seeking sufficient conditions, a task that would be too difficult or impossible due to its deep philosophical complications as well as the logical and technical ones.

Specifically, it is widely known that correlation is not causation. As Granger [62] puts it,

*when discussing the interpretation of a correlation coefficient or a regression, most textbooks warn that an observed relationship does not allow one to say anything about causation between the variables.*

Perhaps that is the reason why Suppes [61] uses the term "prima facie cause" in his definition given above which, however, he does not explain, apart for attributing "prima facie" to Jaakko Hintikka. Furthermore, Suppes discusses *spurious causes* and eventually defines the *genuine cause* as a "prima facie cause that is not spurious"; he also discusses the very existence of genuine causes which under certain conditions (e.g., in a Laplacean universe) seems doubtful.

Granger himself also uses the term "prima facie cause", while Granger and Newbold [63] note that a cause satisfying a causality test still remains prima facie because it is always possible that, if a different information set were used, it would fail the new test. Despite the caution issued by its pioneers, including Granger, through the years, the term "Granger causality" has become popular (particularly in the so-called "Granger causality test", e.g., [64]). Probably because of that misleading term, the technique is sometimes thought of as one that establishes causality, thus resolving or overcoming the "correlation is not causation" problem. In general, it has rarely been understood that identifying genuine causality is not a problem of choosing the best algorithm to establish a statistical relationship (including its directionality) between two variables. As an example of misrepresentation of the actual problems, see [65], which contains the statement:

*Determining true causality requires not only the establishment of a relationship between two variables, but also the far more difficult task of determining a direction of causality.*

In essence, the "Granger causality test" studies the improvement of prediction of a process $\underline{y}_\tau$ by considering the influence of a "causing" process $\underline{x}_\tau$ through the Granger regression model:

$$\underline{y}_\tau = \sum_{j=1}^{\eta} a_j \underline{y}_{\tau-j} + \sum_{j=1}^{\eta} b_j \underline{x}_{\tau-j} + \varepsilon_\tau \tag{14}$$

where $a_j$ and $b_j$ are the regression coefficients and $\varepsilon_\tau$ is an error term. The test is based on the null hypothesis that the process $\underline{x}_\tau$ is not actually causing $\underline{y}_\tau$, formally expressed as

$$H_0 : b_1 = b_2 = \ldots = b_\eta = 0 \tag{15}$$

Algorithmic details of the test are given in [64], among others. The rejection of the null hypothesis is commonly interpreted in the literature with a statement that $\underline{x}_\tau$ "Granger-causes" $\underline{y}_\tau$.

This is clearly a misstatement and, in fact, the entire test is based on correlation matrices. Thus, it again reflects correlation rather than causation. The rejection of the null hypothesis signifies improvement of prediction and this does not mean causation. To make this clearer, let us consider the following example: people sweat when the atmospheric temperature is high and also wear light clothes. Thus, it is reasonably expected that in the prediction of sweat quantity, temperature matters. In the

absence of temperature measurements (e.g., when we have only visual information, like when watching a video), the weight of the clothes algorithmically improves the prediction of sweat quantity. However, we could not say that the decrease in clothes weight causes an increase in sweat (the opposite is more reasonable and becomes evident in a three-variable regression, temperature–clothes weight–sweat, as further detailed in Appendix A.4).

Cohen [66] suggested replacing the term "Granger causality" with "Granger prediction" after correctly pointing out this:

> *Results from Granger causality analyses neither establish nor require causality. Granger causality results do not reveal causal interactions, although they can provide evidence in support of a hypothesis about causal interactions.*

To avoid such philosophical and logical complications, here, we replace the "prima facie" or "Granger" characterization of a cause and, as we already explained, we abandon seeking for genuine causes by using the notion of *necessary conditions* for causality. One could say that if two processes satisfy the necessary conditions, then they define a prima facie causality, but we avoid stressing that as we deem it unnecessary. Furthermore, we drop "causality" from "Granger causality test", thus hereinafter calling it "Granger test".

Some have thought they can approach genuine causes and get rid of the caution "correlation is not causation" by replacing the correlation with other statistics in the mathematical description of causality. For example, Liang [44] uses the concept of information (or entropy) flow (or transfer) between two processes; this method has been called "Liang causality" in the already cited work he co-authors [43]. The usefulness of such endeavours is not questioned, yet their vanity to determine genuine causality is easy to infer: It suffices to consider the case where the two processes, for which causality is studied, are jointly Gaussian. It is well known that in any multivariate Gaussian process, the covariance matrix (or the correlation matrix along with the variances) fully determines all properties of the multivariate distribution of any order. For example, the mutual information in a bivariate Gaussian process is (Papoulis, [67])

$$H[\underline{y}|x] = \ln \sigma_y \sqrt{2\pi e(1 - r^2)} \tag{16}$$

where $\sigma$ and $r$ denote standard deviation and correlation, respectively. Thus, using any quantity related to entropy (equivalently, information), is virtually identical to using correlation. Furthermore, in Gaussian processes, whatever statistic is used in describing causality, it is readily reduced to correlation. This is evident even in Liang [44], where, e.g., in his Equation (102), the information flow turns out to be the correlation coefficient multiplied by a constant. In other words, the big philosophical problem of causality cannot be resolved by technical tricks.

From what was exposed above (Section 4.1), the time irreversibility (or directionality) is most important in seeking causality. In this respect, we certainly embrace Suppes's condition (i) and Granger's first axiom, as stated above. Furthermore, we believe there is no meaning in refusing that axiom and continuing to speak about causality. We note though that there have been recent attempts to show that

> *coupled chaotic dynamical systems violate the first principle of Granger causality that the cause precedes the effect.* [68]

Apparently, however, the particular simulation experiment performed in the latter work which, notably, is not even accompanied by any attempt for deduction based on stochastics, cannot show any violation. In our view, such a violation, if it indeed happened, would be violation of logic and perhaps of common sense.

Additional notes for other procedures detecting causality, which are not included in the focus of our study, are given in Appendix A.4.

### 4.3. Additional Clarifications of Our Approach

After the above theoretical and methodological discourse, we can clarify our methodological approach by emphasizing the following points.

1.  To make our assertions and, in particular, to use the "hen-or-egg" metaphor, we do not rely on merely statistical arguments. If we did that, based on our results presented in the next section, we would conclude that only the causality direction $T \rightarrow [CO_2]$ exists. However, one may perform a thought experiment of instantly adding a big quantity of $CO_2$ to the atmosphere. Would the temperature not increase? We believe it would, as $CO_2$ is known to be a greenhouse gas. The causation in the opposite direction is also valid, as will be discussed in Section 6, "Physical Interpretation". Therefore, we assert that both causality directions exist, and we are looking for the dominant one under the current climate conditions (those manifest in the datasets we use) instead of trying to make assertions of an exclusive causality direction.

2.  While we occasionally use statistical tests (namely, the Granger test, Equations (14) and (15)), we opt to use, as the central point of our analyses, Equation (13) (and the conditions below it) because it is more intuitive and robust, fully reflects the basic causality axiom of time precedence, and is more straightforward, transparent (free of algorithmic manipulations), and easily reproducible (without the need for specialized software).

3.  For simplicity, we do not use any statistic other than correlation here. We stress that the system we are examining is indeed classified as Gaussian and, thus, it is totally unnecessary to examine any statistic in addition to correlation. The evidence of Gaussianity is provided by Figures A1 and A2 in Appendix A.5, in terms of marginal distributions of the processes examined and in terms of their relationship. In particular, Figure A2 suggests a typical linear relationship for the bivariate process. We note that the linearity here is not a simplifying assumption or a coincidence as there are theoretical reasons implying it, which are related to the principle of maximum entropy [67,69].

4.  All in all, we adhere to simplicity and transparency and, in this respect, we illustrate our results graphically, so they are easily understandable, intuitive, and persuasive. Indeed, our findings are easily verifiable even from simple synchronous plots of time series, yet we also include plots of autocorrelations and lagged cross-correlation, which are also most informative in terms of time directionality.

## 5. Results

### 5.1. Original Time Series

Here, we examine the relationship of atmospheric temperature and carbon dioxide concentration using the available modern data (observations rather than proxies) in monthly time steps, as described in Section 3. To apply our stochastic framework, we must first make the two time series linearly compatible. Specifically, based on Arrhenius's rule (Equation (1)), we take the logarithms of $CO_2$ concentration while we keep $T$ untransformed. Such a transformation has also been performed in previous studies, which consider the logarithm of $CO_2$ concentration as a proxy of total radiative forcing (e.g., [41]). However, by calling this quantity "forcing", we indirectly give it, a priori (i.e., before investigating causation), the role of being the cause. Therefore, here, we avoid such interpretations; we simply call this variable the logarithm of carbon dioxide concentration and denote it as $\ln[CO_2]$.

A synchronous plot of the two processes (specifically, UAH temperature and $\ln[CO_2]$ at Mauna Loa) is depicted in Figure 8. Very little can be inferred from this figure alone. Both processes show increasing trends and thus appear as positively correlated. On the other hand, the two processes appear to have different behaviours. Temperature shows an erratic behaviour while $\ln[CO_2]$ has a smooth evolution marked by the annual periodicity. It looks impossible to infer causality from that graph alone.

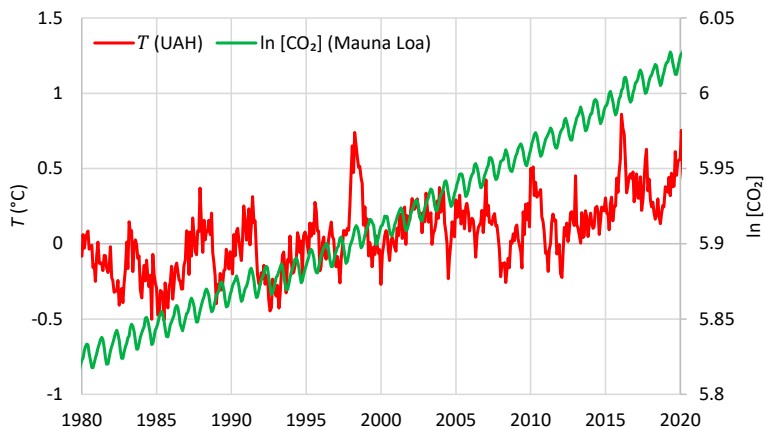

**Figure 8.** Synchronous plots of the time series of UAH temperature and logarithm of $CO_2$ concentration at Mauna Loa at monthly scale.

Somewhat more informative is Figure 9, which depicts lagged cross-correlations of the two processes, based on the methodology in Section 4.1 but without differencing the processes. Specifically, Figure 9 shows the cross-correlogram between UAH temperature and Mauna Loa $\ln[CO_2]$ at monthly and annual scales; the autocorrelograms of the two processes are also plotted for comparison. In both time scales, the cross-correlogram shows high correlations at all lags, with the maximum attained at lag zero. This does not hint at a direction. However, the cross-correlations for negative lags are slightly greater than those in the positive lags. Notice that to make this clearer, we have also plotted the differences $r_j - r_{-j}$ in the graph. This behaviour could be interpreted as supporting the causality direction $[CO_2] \to T$. However, we deem that the entire picture is spurious as it is heavily affected by the fact that the autocorrelations are very high and, in particular, those of $\ln[CO_2]$ are very close to 1 for all lags shown in the figure.

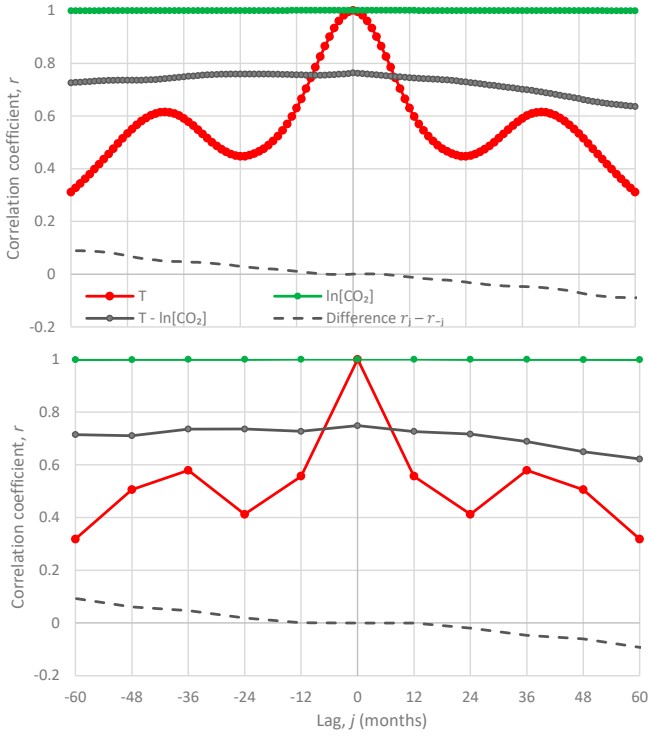

**Figure 9.** Auto- and cross-correlograms of the time series of UAH temperature and logarithm of $CO_2$ concentration at Mauna Loa.

In our investigation, we also applied the Granger test on these two time series in both time directions. To calculate the *p*-value of the Granger test, we used free software (namely the function GRANGER_TEST [70,71]). It appears that in the causality direction $[CO_2] \rightarrow T$, the null hypothesis is rejected at all usual significance levels. The attained *p*-value of the test is $1.8 \times 10^{-7}$ for one regression lag ($\eta = 1$), $1.8 \times 10^{-4}$ for $\eta = 2$, and remains below 0.01 for subsequent $\eta$. By contrast, in the direction $T \rightarrow [CO_2]$, the null hypothesis is not rejected at all usual significance levels. The attained *p*-value of the test is 0.25 for $\eta = 1$, 0.22 for $\eta = 2$, and remains above 0.1 for subsequent $\eta$.

Therefore, one could directly interpret these results as unambiguously showing one-way causality between the total greenhouse gases and temperature and, hence, validating the consensus view that human activity is responsible for the observed rise in global temperature. However, these results are certainly not unambiguous and, most probably, they are spurious. To demonstrate that they are not unambiguous, we have plotted, as shown in the upper panels of Figure 10, the *p*-values of the Granger test for moving windows with a size of 10 years for number of lags $\eta = 1$ and 2. The values for the entire length of time series, as given above, are also shown as dashed lines. Now the picture is quite different: each of the two directions appear dominating (meaning that the attained significance level is lower in one over the other) in about equal portions of the time. For example, for $\eta = 2$, the $T \rightarrow [CO_2]$ dominates over $[CO_2] \rightarrow T$ for 58% of the time. The attained *p*-value for direction $T \rightarrow [CO_2]$ is lower than 1% for 1.4% of the time, much higher than in the opposite direction (0.3% of the time). All of these observations favour the $T \rightarrow [CO_2]$ direction.

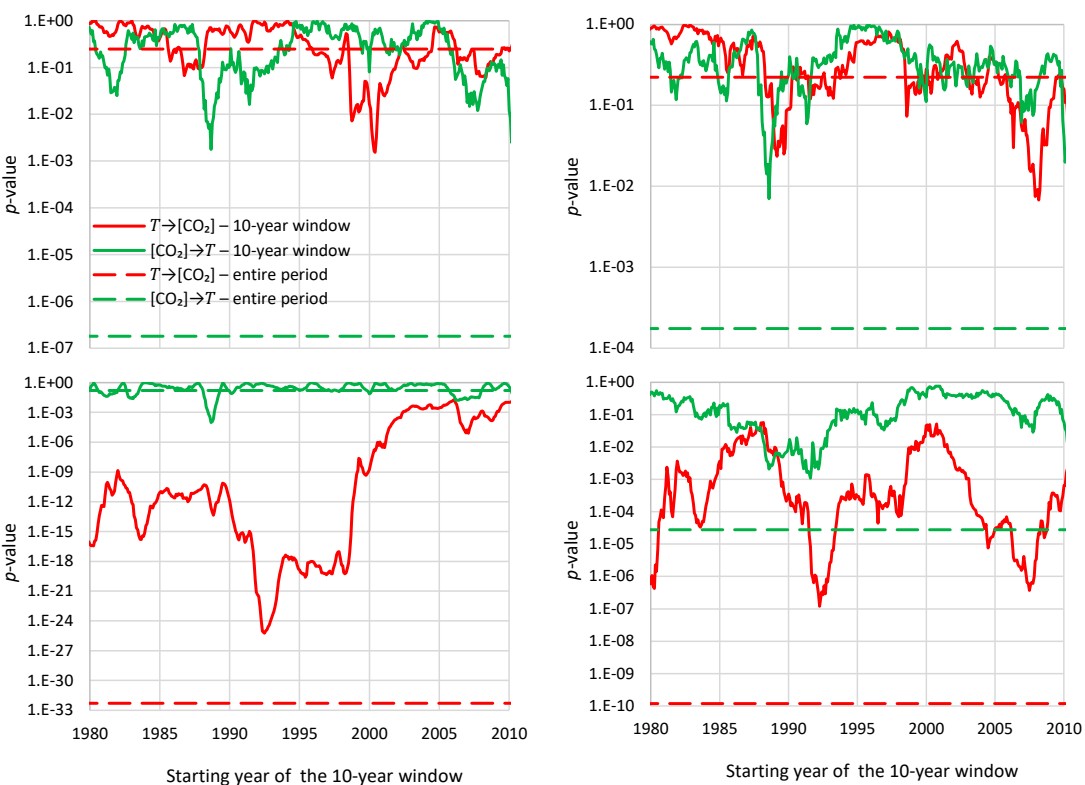

**Figure 10.** Plots of *p*-values of the Granger test for 10-year-long moving windows for the monthly time series of UAH temperature and logarithm of $CO_2$ concentration at Mauna Loa for number of lags (**left**) $\eta = 1$ and (**right**) $\eta = 2$. The time series used are (**upper**) the original and (**lower**) that obtained after "removing" the periodicity by averaging over the previous 12 months.

To show that the results are spurious and, in particular, affected by the very high autocorrelations of $\ln[CO_2]$ and, more importantly, by its annual cyclicity, we have "removed" the latter by averaging over the previous 12 months. We did that for both series and plotted the results in the lower panels

of Figure 10. Here, the results are stunning. For both lags $\eta = 1$ and 2 and for the entire period (or almost), $T \rightarrow [CO_2]$ dominates, attaining $p$-values as low as in the order of $10^{-33}$. However, we will avoid interpreting these results as unambiguous evidence that the consensus view (i.e., human activity is responsible for the observed warming) is wrong. Rather, what we want to stress is that it is inappropriate to draw conclusions from a methodology which is demonstrated to be so sensitive to the used time windows and data processing assumptions. In this respect, we have included this analyses in our study only (a) to show its weaknesses (which, for the reasons we explained in Section 4.2, we believe would not change if we used different statistics or different time series) and (b) to connect our study to earlier ones. For the sake of drawing conclusions, we contend that our full methodology in Sections 4.1 and 4.3 is more appropriate. We apply this methodology in Section 5.2.

### 5.2. Differenced Time Series

We have already explained the advantages of investigating the differenced processes, which quantify changes from a mathematical and logical point of view. In our case, taking differences is also physically meaningful as both $CO_2$ concentration and temperature (equivalent to thermal energy) represent "stocks", i.e., stored quantities and, thus, the mass and energy fluxes are indeed represented by differences.

The chosen time step of differencing is equal to one year ($\nu = 12$ for the monthly time step of the time series). For instance, from the value of January of a certain year, we subtract the value of January of the previous year and so forth. A first reason for this choice is that it almost eliminates the effect of the annual cycle (periodicity). A second reason is that the temperature data are given in terms of "anomalies", i.e., differences from an average which varies from month to month. By taking $\nu = 12$, the varying means are eliminated, and "anomalies" are effectively replaced by the actual processes (as the differences in the actual values equal the differences of "anomalies").

We perform all analyses on both monthly and annual time scales. Figure 11 shows the differenced time series for the UAH temperature and Mauna Loa $CO_2$ concentration at monthly scale; the symbols $\Delta T$ and $\Delta \ln[CO_2]$ are used interchangeably with $\underline{\tilde{x}}_{\tau,12}$ and $\underline{\tilde{y}}_{\tau,12}$, respectively.

Comparing Figure 8 (undifferenced series) with Figure 11 (differenced series), one can verify that the latter is much more informative in terms of the directionality of the relationship of the two processes. While Figure 8 did not provide any relevant hints, Figure 11 clearly shows that, most often, the temperature curve leads and that of $CO_2$ follows. However, there are cases where the changes in the two processes synchronize in time or even become decoupled.

Figure 12 shows the same time series at the annual time scale, with the year being defined as July–June for $\Delta T$ and February–January for $\Delta \ln[CO_2]$. The reason for this differentiation will be explained below. Here, it is more evident that, most of the time, the temperature change leads and that of $CO_2$ follows.

It is of interest here that the variability of global mean annual temperature is significantly influenced by the rhythm of ocean–atmosphere oscillations, such as ENSO, AMO, and IPO [72]. This mechanism may be a complicating factor, in turn influencing the link between temperature and $CO_2$ concentration. This is not examined here (except a short note in the end of the section) as, given the focus in examining just the connection of the latter two processes, it lies out of our present scope.

The climacograms of the differenced time series used (actually four of the six to avoid an overcrowded graph) are shown in Figure 13. It appears that the differenced temperature time series are consistent with the condition implied by stationarity, i.e., $H = 0$ for the differenced process. The same does not look to be the case for the $CO_2$ time series, particularly for the Mauna Loa time series, in which the Hurst parameter appears to be close to 1/2. Based on this, one would exclude stationarity for the Mauna Loa $CO_2$ time series. However, a simpler interpretation of the graph is that the data record is not long enough to reveal that $H = 0$ for the differenced process. Actually, all available data belong to a period in which $[CO_2]$ exhibits a monotonic increasing trend (as also verified by the fact that all values of $\Delta \ln[CO_2]$ in Figures 11 and 12 are positive, while stationarity entails a zero mean of the differenced

process). Had the available database been broader, both positive and negative trends could appear. Indeed, a broader view of the [CO₂] process based on palaeoclimatic data (Figures 3 and 4) would justify a stationarity assumption.

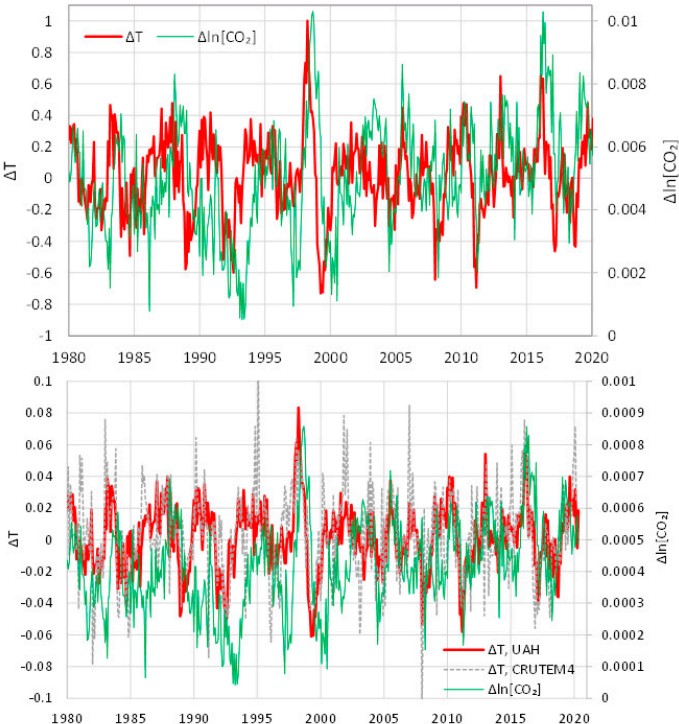

**Figure 11.** Differenced time series of UAH temperature and logarithm of $CO_2$ concentration at Mauna Loa at monthly scale. The graph in the upper panel was constructed in the manner described in the text. The graph in the lower panel is given for comparison and was constructed differently by taking differences of the values of each month with the previous month and then averaging over the previous 12 months (to remove periodicity); in addition, the lower graph includes the CRUTEM4 land temperature series.

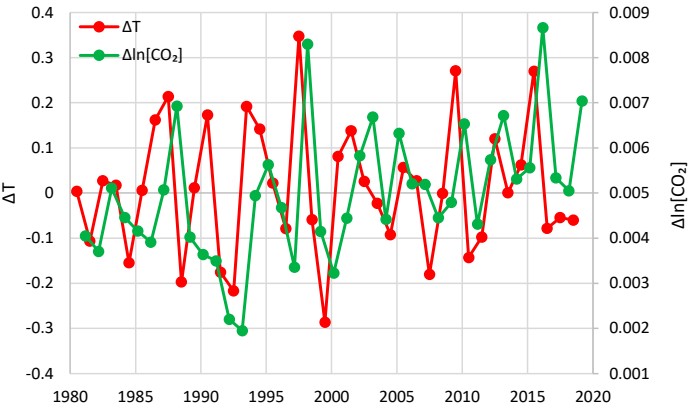

**Figure 12.** Annually averaged time series of differenced temperatures (UAH) and logarithms of $CO_2$ concentrations (Mauna Loa). Each dot represents the average of a one-year duration ending at the time of its abscissa.

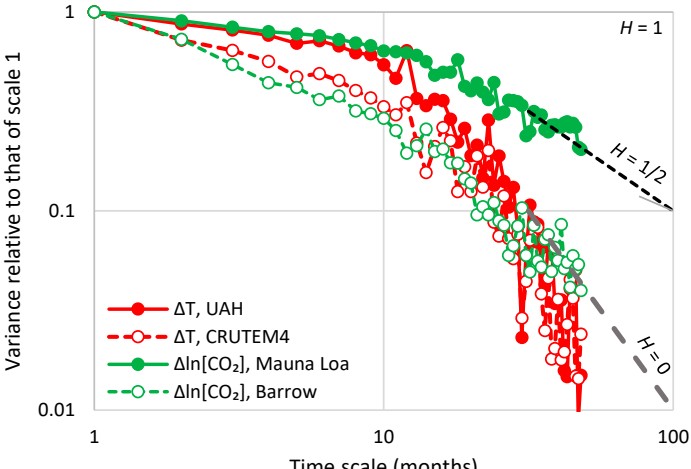

**Figure 13.** Empirical climacograms of the indicated differenced time series; the characteristic slopes corresponding to values of the Hurst parameter $H = 1/2$ (large-scale randomness), 0 (full antipersistence) and 1 (full persistence) are also plotted (note, $H = 1 +$ slope/2).

The preliminary qualitative observation from graphical inspection of Figures 11 and 12 suggests that the temperature change very often precedes and the $CO_2$ change follows in the same direction. We note, though, that temperature changes alternate in sign while $CO_2$ changes are always positive.

A quantitative analysis based on the methodology in Section 4.1 requires the study of lagged cross-correlations of the two processes. Figure 14 shows the cross-correlogram between UAH temperature and Mauna Loa $CO_2$ concentration; the autocorrelograms of the two processes are also plotted for comparison. The fact that the cross-correlogram does not have values consistently close to zero at any of the semi-axes eliminates the possibility of an exclusive (unidirectional) causality and suggests consistency with "hen-or-egg" causality.

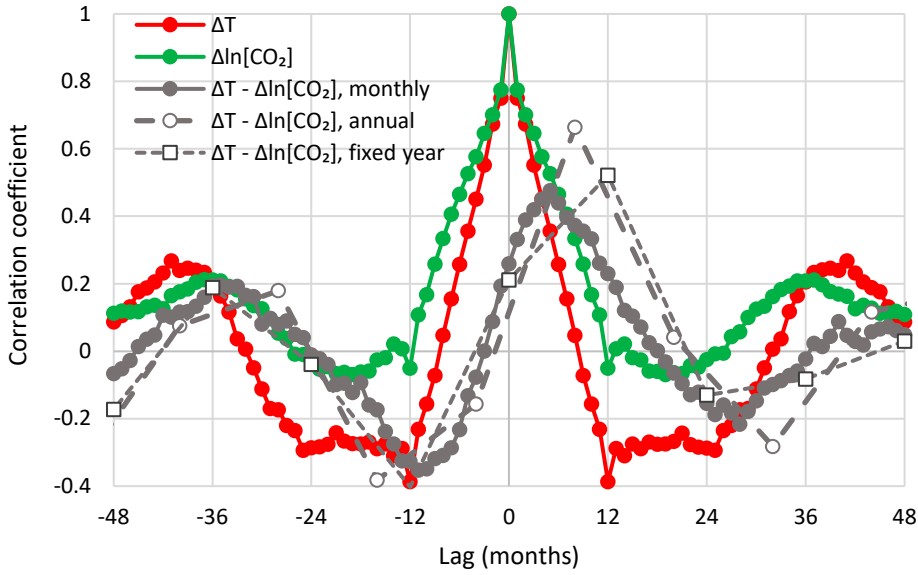

**Figure 14.** Auto- and cross-correlograms of the differenced time series of UAH temperature and Mauna Loa $CO_2$ concentration.

The maximum cross-correlation of the monthly series is 0.47 and appears at a positive lag, $\eta_1 = 5$ months, thus suggesting $T \rightarrow [CO_2]$, rather than $[CO_2] \rightarrow T$, as dominant causality direction. Similar

are the graphs of the other combinations of temperature and $CO_2$ datasets, which are shown in Appendix A.5 (Figures A3–A7). In all cases, $\eta_1$ is positive, ranging from 5 to 11 months.

To perform similar analyses on the annual scale, we fixed the specification of a year for temperature for the period July–June, as already mentioned, and then slid the initial month specifying the beginning of a year for $CO_2$ concentration so as to find a specification that maximizes the cross-correlation at the annual scale. In Figure 14, maximization occurs when the year specification is February–January (of the next year), i.e., if the lag is 8 months. The maximum cross-correlation is 0.66. If we keep the specification of the year for $CO_2$ concentration the same as in temperature (July–June), then maximization occurs at lag one year (12 months) and the maximum cross-correlation is 0.52. Table 1 summarizes the results for all combinations examined. The lags are always positive. They vary between 8 and 14 months for a sliding window specification and are 12 months for the fixed window specification. Most interestingly, the opposite phase in the annual cycle of $CO_2$ concentration in the South Pole, with respect to the other three sites, does not produce any noteworthy differences in the shape of the cross-correlogram nor in the time lags maximizing the cross-correlations.

**Table 1.** Maximum cross-correlation coefficient (MCCC) and corresponding time lag in months. The annual window for temperature is July–June, while for $CO_2$, it is either different (sliding), determined so as to maximize MCCC, or the same (fixed).

| Temperature—$CO_2$ Series | Monthly Time Series | | Annual Time Series—Sliding Annual Window | | Annual Time Series—Fixed Annual Window | |
|---|---|---|---|---|---|---|
| | MCCC | Lag | MCCC | Lag | MCCC | Lag |
| UAH—Mauna Loa | 0.47 | 5 | 0.66 | 8 | 0.52 | 12 |
| UAH—Barrow | 0.31 | 11 | 0.70 | 14 | 0.59 | 12 |
| UAH—South Pole | 0.37 | 6 | 0.54 | 10 | 0.38 | 12 |
| UAH—Global | 0.47 | 6 | 0.60 | 11 | 0.60 | 12 |
| CRUTEM4—Mauna Loa | 0.31 | 5 | 0.55 | 10 | 0.52 | 12 |
| CRUTEM4—Global | 0.33 | 9 | 0.55 | 12 | 0.55 | 12 |

While, as explained in Sections 4.2 and 5.1, the Granger test has weaknesses that may not help in drawing conclusions, for completeness and as a confirmation, we list its results here:

- For the monthly scale and the causality direction $[CO_2] \rightarrow T$, the null hypothesis is not rejected at all usual significance levels for lag $\eta = 1$ and is rejected for significance level 1% for $\eta = 2–8$, with minimum attained $p$-value $1.8 \times 10^{-4}$ for $\eta = 6$.
- For the monthly scale and the causality direction $T \rightarrow [CO_2]$, the null hypothesis is rejected at all usual significance levels for all lags $\eta$, with minimum attained $p$-value $2.1 \times 10^{-8}$ for $\eta = 7$.
- For the monthly scale, the attained $p$-values in the direction $T \rightarrow [CO_2]$ are always smaller than in direction $[CO_2] \rightarrow T$ by about 4 to 5 orders of magnitude, thus clearly supporting $T \rightarrow [CO_2]$ as dominant direction.
- For the annual scale with fixed year specification and the causality direction $[CO_2] \rightarrow T$, the null hypothesis is not rejected at all usual significance levels for any lag $\eta$, thus indicating that this causality direction does not exist.
- For the annual scale with fixed year specification and the causality direction $T \rightarrow [CO_2]$, the null hypothesis is not rejected at significance level 1% for all lags $\eta = 1–6$, with minimum attained $p$-value 5% for lag $\eta = 2$, thus supporting this causality direction at this significance level.
- For the annual scale with fixed year specification, the attained $p$-values in the direction $T \rightarrow [CO_2]$ are always smaller than in direction $[CO_2] \rightarrow T$, again clearly supporting $T \rightarrow [CO_2]$ as the dominant direction.

We note that the test cannot be applied for the sliding time window case and, hence, we cannot provide results for this case.

We add a final remark, in view of a comment by Masters and Benestad [73] on the already mentioned study by Humlum et al. [45], in which they claim that "the inter-annual fluctuations in atmospheric $CO_2$ produced by ENSO can lead to a misdiagnosis of the long-term cause of the recent atmospheric $CO_2$ increase". Inspired by this comment, we have made a preliminary three-variable investigation using differenced temperatures (UAH), logarithms of $CO_2$ concentrations (Mauna Loa), and Equatorial South Oscillation index (SOI) characterizing ENSO. The investigation has been made on a monthly scale. $\Delta \ln[CO_2]$ has been linearly regressed with $\Delta T$ and the running average of SOI for the previous 12 months. At synchrony (without applying any time lag), the correlation of SOI with $\Delta T$ is 0.40, higher than that of $\Delta T$ and $\Delta \ln[CO_2]$ (0.24, as seen in Figure 14 at lag 0). The highest determination coefficient for the three regressed quantities is obtained when the time lag between $\Delta \ln[CO_2]$ and $\Delta T$ is again 5 months, as in the two-variable case (the optimal lag for SOI is 0, but the regression is virtually insensitive to the change of that lag). Its value is $r^2 = 0.23$, corresponding to $r = 0.48$, i.e., only slightly higher than the maximum cross-correlation coefficient of the two variable-case (which is 0.47 as seen in Table 1). In other words, by including ENSO in the modelling framework, the results do not change.

In brief, all above confirm the results of our methodology that the dominant direction of causality is $T \rightarrow [CO_2]$.

## 6. Physical Interpretation

The omnipresence of positive lags on both monthly and annual time scales and the confirmation by Granger tests reduce the likelihood that our results are statistical artefacts. Still, our results require physical interpretation which we seek in the natural process of soil respiration.

Soil respiration, $R_s$, defined to be the flux of microbially and plant-respired $CO_2$, clearly increases with temperature. It is known to have increased in the recent years [74,75]. Observational data of $R_s$ (e.g., [76,77]; see also [78]) show that the process intensity increases with temperature. Rate of chemical reactions, metabolic rate, as well as microorganism activity, generally increase with temperature. This has been known for more than 70 years (Pomeroy and Bowlus [79]) and is routinely used in engineering design.

The Figure 6.1 of the latest report of the IPCC [75] provides a quantification of the mass balance of the carbon cycle in the atmosphere that is representative of recent years. The soil respiration, assumed to be the sum of respiration (plants) and decay (microbes), is 113.7 Gt C/year (IPCC gives a value of 118.7 including fire, which along with biomass burning, is estimated to be 5 Gt C/year by Green and Byrne [80]).

We can expect that sea respiration would also have increased. Moreover, outgassing from the oceans must also have increased as the solubility of $CO_2$ in water decreases with increasing temperature [14,81]. In addition, photosynthesis must have increased, as in the 21st century the Earth has been greening, mostly due to $CO_2$ fertilization effects [82] and human land-use management [83]. Specifically, satellite data show a net increase in leaf area of 2.3% per decade [83]. The sums of carbon outflows from the atmosphere (terrestrial and maritime photosynthesis as well as maritime absorption) amount to 203 Gt C/year. The carbon inflows to the atmosphere amount to 207.4 Gt C/year and include natural terrestrial processes (respiration, decay, fire, freshwater outgassing as well as volcanism and weathering), natural maritime processes (respiration) as well as anthropogenic processes. The latter comprise human $CO_2$ emissions related to fossil fuels and cement production as well as land-use change, and amount to 7.7 and 1.1 Gt C/year, respectively. The change in carbon fluxes due to natural processes is likely to exceed the change due to anthropogenic $CO_2$ emissions, even though the latter are generally regarded as responsible for the imbalance of carbon in the atmosphere.

## 7. Conclusions

The relationship between atmospheric concentration of carbon dioxide and the global temperature is widely recognized, and it is common knowledge that increasing $CO_2$ concentration plays a major role in enhancement of the greenhouse effect and contributes to global warming.

While the fact that these two variables are tightly connected is beyond doubt, the direction of the causal relationship needs to be studied further. The purpose of this study is to complement the conventional and established theory, that increased $CO_2$ concentration due to anthropogenic emissions causes an increase of temperature, by considering the concept of reverse causality. The problem is obviously more complex than that of exclusive roles of cause and effect, qualifying as a "hen-or-egg" ("ὄρνις ἢ ᾠὸν") causality problem, where it is not always clear which of two interrelated events is the cause and which the effect. Increased temperature causes an increase in $CO_2$ concentration and, hence, we propose the formulation of the entire process in terms of a "hen-or-egg" causality.

We examine the relationship of global temperature and atmospheric carbon dioxide concentration using the most reliable global data that are available—the data gathered from several sources, covering the common time interval 1980–2019, available at the monthly time step.

The results of the study support the hypothesis that both causality directions exist, with $T \to CO_2$ being the dominant, despite the fact that $CO_2 \to T$ prevails in public, as well as in scientific, perception. Indeed, our results show that changes in $CO_2$ follow changes in $T$ by about six months on a monthly scale, or about one year on an annual scale.

The opposite causality direction opens a nurturing interpretation question. We attempted to interpret this mechanism by noting that the increase in soil respiration, reflecting the fact that the intensity of biochemical process increases with temperature, leads to increasing natural $CO_2$ emission. Thus, the synchrony of rising temperature and $CO_2$ creates a positive feedback loop. This poses challenging scientific questions of interpretation and modelling for further studies. In this respect, we welcome the review by Connolly [14], which already proposes interesting interpretations within a wider epistemological framework and in connection with a recent study [84]. In our opinion, scientists of the 21st century should have been familiar with unanswered scientific questions as well as with the idea that complex systems resist simplistic explanations.

**Author Contributions:** Conceptualization, D.K.; methodology, D.K.; software: D.K.; validation, Z.W.K.; formal analysis, D.K.; investigation, D.K. and Z.W.K.; data curation, D.K.; writing—original draft preparation, D.K. and Z.W.K.; writing—review and editing, D.K. and Z.W.K.; visualization, D.K. and Z.W.K. All authors have read and agreed to the published version of the manuscript.

**Funding:** This research received no external funding but was motivated by the scientific curiosity of the authors.

**Acknowledgments:** Some negative comments of two anonymous reviewers for an earlier submission of this manuscript in another journal (which we have posted online: manuscript at http://dx.doi.org/10.13140/RG.2.2.29154.15045/1, reviews at http://dx.doi.org/10.13140/RG.2.2.14524.87681) helped us improve the presentation and strengthen our arguments against their comments. We appreciate these reviewers' suggestions of relevant published works, which we were unaware of. We are grateful to the reviewers of the submission of our study to *Sci*, Yog Aryal, Ronan Connolly, and Stavros Alexandris, whose constructive comments helped us to improve the paper, mostly by adding a lot of additional information and clarification in appendices. The discussion of DK with Antonis Christofides helped to substantially improve an initial version of Appendix A.4, which he informally reviewed. We thank the journal *Sci* for the interesting experience it offered us.

**Conflicts of Interest:** The authors declare no conflict of interest.

**Data Availability:** The two temperature time series and the Mauna Loa $CO_2$ time series are readily available on monthly scale from http://climexp.knmi.nl. All NOAA $CO_2$ data are available from https://www.esrl.noaa.gov/gmd/ccgg/trends/gl_trend.html. The $CO_2$ data of Mauna Loa were retrieved from http://climexp.knmi.nl/data/imaunaloa_f.dat while the original measurements are in https://www.esrl.noaa.gov/gmd/dv/iadv/graph.php?code=MLO. The Barrow series is available (in irregular step) in https://www.esrl.noaa.gov/gmd/dv/iadv/graph.php?code=BRW, and the South Pole series in https://www.esrl.noaa.gov/gmd/dv/data/index.php?site=SPO. All these data were accessed (using the "Download data" link in the above sites) in June 2020. The global $CO_2$ series is accessed at https://www.esrl.noaa.gov/gmd/ccgg/trends/gl_data.html, of which the "Globally averaged marine surface monthly mean data" are used here. The palaeoclimatic data of Vostok $CO_2$ were retrieved from http://cdiac.ess-dive.lbl.gov/ftp/trends/co2/vostok.icecore.co2 (dated January 2003, accessed September 2018) and the temperature data from http://cdiac.ess-dive.lbl.gov/ftp/trends/temp/vostok/vostok.1999.temp.dat (dated January 2000, accessed September 2018).

## Appendix A

*Appendix A.1. On Early Non-Systematic Measurements of CO$_2$*

This Appendix (not contained in Version 1 of our paper) addresses comments by all three reviewers of Version 1, Yog Aryal [85], Ronan Connolly [14], and Stavros Alexandris [86], about the reasons why we delimit our analysis to the period 1980–2019. The two latter reviewers suggested using earlier data compiled by Beck (2007), who referred to old chemical analyses of atmospheric concentration of CO$_2$.

We are sympathetic to the passion of the late Ernst-Georg Beck who, being a biology teacher, sacrificed a lot of time and effort to the exciting exercise of digging out old CO$_2$ measurements. Indeed, it could be worthwhile to have a critical look at the historical data and to try to make order in them and utilize them. However, this would certainly warrant an individual paper with this particular aim.

Historically, it was not the first review paper of this sort. For instance, in his Table 1, Beck [87] refers to old works by Letts and Blake (~1900; [88]), who considered 252 papers with data (all in 19th century), and to Stepanova [89], who considered 229 papers with data (130 in 19th century and 99 in 20th century). Beck himself [87] considered 156 papers with data (82 in 19th and 74 in 20th century).

As usual, it is instructive to consider the paper by Beck [87] jointly with critical commentaries published later in the journal where the original paper appeared [90,91]. In particular, R.F. Keeling [90] opined that the old chemical measurements examined by Beck [87] "exhibit far too much geographic and short-term temporal variability to plausibly be representative of the background. The variability of these early measurements must therefore be attributed to 'local or regional' factors or poor measurement practice". Keeling [90] also noted "basic accounting problems". "Beck's 11-year averages show large swings, including an increase from 310 to 420 ppm between 1920 and 1945 (Beck's Figure 11)". "To drive an increase of this magnitude globally requires the release of 233 billion metric tons of carbon to the atmosphere. The amount is equivalent to more than a third of all the carbon contained in land plants globally. [ ... ] To make a credible case, Beck needed to offer evidence for losses or gains of carbon of this magnitude from somewhere. He offered none."

Meijer [91] expressed the opinion that Beck's work "contains major flaws, such that the conclusions are wrong". He also wrote: "The measurements presented in the paper are indeed useless for the purpose the author wants to use them, certainly in the way the author interprets them". There is a lack of interpretation of diurnal and seasonal variability (effects called the "diurnal" and the "seasonal" rectifier in the literature) and consideration of atmospheric mixing or lack thereof. Meijer also criticized the lack of meta-data: "The necessary data to judge, namely measurement height, consecutive length of a record and especially temporal resolution, are lacking in [Beck's] Table 2. In the light of the above, the whole 'Discussion and Conclusion' section is invalid, including [Beck's] Figures 11–14". Indeed, the records mentioned in Beck's Table 2 were local and short-lasting, with the longest periods being 1920–1926. Beck's Figures 11 and 13 show concatenated short segments of data at different places.

There are some other puzzling elements in Beck's paper. For instance, in his Figure 5, referring to data from a meteorological station near Giessen, the variability of high amplitude seems suspicious and not physically realistic. In particular, from June to August 1940, the measured CO$_2$ concentration increases from 340 to 550 ppm (much more than in Beck's Figure 5 discussed by Keeling [90] and Meijer [91] as quoted above), with weird seasonal behaviour. Beck himself admitted that the results for Giessen "need to be adjusted downwards to take account of anthropogenic sources of CO$_2$ from nearby city, an influence that has been estimated as lying between 10 and 70 ppm [ ... ] by different authors".

The controversy and disputes among these authors extended beyond pure scientific issues. Thus, Beck [87] wrote "[t]he data accepted [ ... ] had to be sufficiently low to be consistent with the greenhouse hypothesis of climate change controlled by rising CO$_2$ emissions from fossil fuel burning". On the other hand, Meijer [91] wrote: "The author even accuses the pioneers Callendar and [Charles David] Keeling of selective data use, errors or even something close to data manipulation". In addition, [R.F.] Keeling [90] noted: "Beck is [ ... ] wrong when he asserts that the earlier data have been discredited only because they don't fit a preconceived hypothesis of CO$_2$ and climate. [ ... ] Instead, the data have

been ignored because they cannot be accepted as representative without violating our understanding of how fast the atmosphere mixes".

In view of the above questions about the data reliability as well as the controversies and disputes, we decided to limit the period of our study in 1980–2019 in which the measurements are systematic and verifiable because they are made in several locations simultaneously.

*Appendix A.2. Some Notes on the Averaged Differenced Process*

The cumulative process of the differenced process $\underline{\tilde{x}}_{\tau,\nu}$ will be

$$\underline{\tilde{X}}_{\kappa,\nu} := \underline{\tilde{x}}_{1,\nu} + \underline{\tilde{x}}_{2,\nu} + \ldots + \underline{\tilde{x}}_{\kappa,\nu} = \underline{x}_{1+\nu} - \underline{x}_1 + \underline{x}_{2+\eta} - \underline{x}_2 + \ldots + \underline{x}_{\kappa+\nu} - \underline{x}_\kappa$$
$$= \underline{X}_{\kappa+\nu} - \underline{X}_\nu - \underline{X}_\kappa. \tag{A1}$$

Note that for $\eta = 1$, this simplifies to

$$\underline{\tilde{X}}_{\kappa,1} = \underline{X}_{\kappa+1} - \underline{X}_1 - \underline{X}_\kappa = \underline{x}_{\kappa+1} - \underline{x}_1 = \underline{\tilde{x}}_{\kappa,1} =: \underline{\tilde{x}}_\kappa. \tag{A2}$$

Following Equation (7), the average differenced process at discrete time scale $\kappa = \eta$ will be

$$\underline{\tilde{x}}_\tau^{(\kappa)} = \frac{\underline{\tilde{X}}_{\tau\kappa,\kappa} - \underline{\tilde{X}}_{(\tau-1)\kappa,\kappa}}{\kappa} = \frac{(\underline{X}_{\tau\kappa+\kappa} - \underline{X}_\kappa - \underline{X}_{\tau\kappa}) - (\underline{X}_{(\tau-1)\kappa+\kappa} - \underline{X}_\kappa - \underline{X}_{(\tau-1)\kappa})}{\kappa} \tag{A3}$$

which, noting that in the rightmost part, the two terms $\underline{X}_\kappa$ cancel each other and by virtue of (7), simplifies to

$$\underline{\tilde{x}}_\tau^{(\kappa)} = \underline{x}_{\tau+1}^{(\kappa)} - \underline{x}_\tau^{(\kappa)} = \underline{\tilde{x}}_{\tau,1}^{(\kappa)}. \tag{A4}$$

In other words, the average differenced process equals the differenced average process in the case that the differencing time step $\eta$ has been chosen as equal to the averaging time scale $\kappa$. For $\kappa = \eta = 1$, we have $\underline{\tilde{x}}_\tau^{(1)} \equiv \underline{\tilde{x}}_{\tau,1} \equiv \underline{\tilde{x}}_\tau$.

*Appendix A.3. Some Notes on Time Directionality of Causal Systems*

In a unidirectional causal system in continuous time $t$, in which the process $\underline{x}(t)$ is the cause of $\underline{y}(t)$, an equation of the form

$$\underline{y}(t) = \int_0^\infty \alpha(s)\underline{x}(t-s)\mathrm{d}s \tag{A5}$$

should hold [67], where $\alpha(t)$ is the impulse response function. The causality condition is thus

$$\alpha(t) = 0 \text{ for } t < 0. \tag{A6}$$

Here, we consider systems with positive dependence, in which $\alpha(t) \geq 0$ for $t \geq 0$, which are possibly also excited by another process $\underline{v}(t)$, independent of $\underline{x}(t)$. Working in discrete time, we write

$$\underline{y}_\tau = \sum_{j=0}^\infty \alpha_j \underline{x}_{\tau-j} + \underline{v}_\tau. \tag{A7}$$

Assuming (without loss of generality) zero means for all processes, multiplying by $\underline{x}_{\tau-\eta}$, taking expected values, and denoting the cross-covariance function as $c_{xy}[\eta] := \mathrm{E}\left[\underline{x}_{\tau-\eta}\underline{y}_\tau\right]$ and the autocovariance function as $c_x[\eta] := \mathrm{E}\left[\underline{x}_{\tau-\eta}\underline{x}_\tau\right]$, we find

$$c_{xy}[\eta] = \sum_{j=0}^\infty \alpha_j c_x[\eta - j]. \tag{A8}$$

For $\eta > 0$, using the property that $c_x[\eta]$ is an even function ($c_x[\eta] = c_x[-\eta]$), we get

$$c_{xy}[\eta] = \sum_{j=0}^{\infty} \alpha_j c_x[j - \eta] = \sum_{j=0}^{\eta-1} \alpha_j c_x[\eta - j] + \sum_{j=\eta}^{\infty} \alpha_j c_x[j - \eta], \tag{A9}$$

and for the negative part

$$c_{xy}[-\eta] = \sum_{j=0}^{\infty} \alpha_j c_x[j + \eta]. \tag{A10}$$

With intuitive reasoning, assuming that the autocovariance function is decreasing ($c_x[j'] < c_x[j]$ for $j' > j$), as usually happens in natural processes, we may see that the rightmost term of Equations (A9) and (A10) should be decreasing functions of $\eta$ (as for $j' > j$ it will be $c_x[j' - \eta] < c_x[j - \eta]$ and $c_x[j' + \eta] < c_x[j + \eta]$). However, the term $\sum_{j=0}^{\eta-1} \alpha_j c_x[\eta - j]$ of Equation (A9) is not decreasing. Therefore, it should attain a maximum value at some positive lag $\eta = \eta_1$. Thus, a positive maximizing lag, $\eta = \eta_1 > 0$, is a necessary condition for causality direction from $\underline{x}_\tau$ to $\underline{y}_\tau$. Conversely, the condition that the maximizing lag is negative is a sufficient condition to exclude the causality direction exclusively from $\underline{x}_\tau$ to $\underline{y}_\tau$.

All above arguments remain valid if we standardize (divide) by the product of standard deviations of the processes $\underline{x}_\tau$ and $\underline{y}_\tau$ and, thus, we can replace cross-covariances $c_{xy}[\eta]$ with cross-correlations $r_{xy}[\eta]$ (or, in the case of differenced processes, $r_{\tilde{x}\tilde{y}}[\nu, \eta]$).

*Appendix A.4. Some Notes on the Alternative Procedures on Causality*

Reviewer Yog Aryal [85] opined that we missed referring to the recent relevant works by Hannart et al. [92] and Verbitsky et al. [93]. In response to this comment, we include this Appendix (not contained in Version 1 of our paper) explaining, in brief, why we do not compare our results with the ones of those studies, also noting that only the latter study contains material that is prima facie comparable to ours. The former study, focusing on the so-called causal counterfactual theory, is more theoretical and also much more interesting. While we, too, are preparing a theoretical study, in which we will discuss some theories in detail, in this Appendix, we give some key elements of our theoretical disagreements and a counterexample that illustrates the disagreements.

We first note that in order to define causality, Hannart et al. [92] refer to the work on the 18th century philosopher David Hume and, in particular, his famous book *Enquiry concerning Human Understanding* [94] first published in 1748. From this book, we wish to quote the following important passage, which emphasizes the difficulties even in defining causality:

> *Our thoughts and enquiries are, therefore, every moment, employed about this relation: Yet so imperfect are the ideas which we form concerning it, that it is impossible to give any just definition of cause, except what is drawn from something extraneous and foreign to it.*

Hannart et al. [92], while studying the probability of occurrence of an event $Y$, introduced the two-valued variable $X_f$ to indicate whether or not a forcing $f$ is present, and continue as follows:

> *The probability $p_1 = P(Y = 1 | X_f = 1)$ of the event occurring in the real world, with f present, is referred to as factual, while $p_0 = P(Y = 1 | X_f = 0)$ is referred to as counterfactual. Both terms will become clear in the light of what immediately follows. The so-called fraction of attributable risk (FAR) is then defined as*
>
> $$\text{FAR} = 1 - \frac{p_0}{p_1} \tag{A11}$$
>
> *The FAR is interpreted as the fraction of the likelihood of an event that is attributable to the external forcing.*

They also show that under some conditions, FAR is a probability which they denote PN and call probability of necessary causality. They stress that it "is important to distinguish between necessary and sufficient causality" and they associate PN (or FAR) "with the first facet of causality, that of

necessity". They claim to have "introduced its second facet, that of sufficiency, which is associated with the symmetric quantity $1 - (1 - p_1)/(1 - p_0)$"; they denote it as PS, standing for probability of sufficient causality.

Central to the logical framework of Hannart et al. [92] is the notion of *intervention* of an experimenter, which is equivalent to experimentation with the ability to set the value of the assumed cause to a desired value. Clearly, this is feasible in laboratory experiments and infeasible in natural processes. The authors resort to the "so-called *in silico experimentation*" which, despite the impressive name chosen, is intervention in a mathematical model that represents the process. Hence, objectively, they examine the "causality" that is embedded in the model rather than the natural causality. One may argue that this it totally unnecessary. It would be better to inspect the model's equations or code to investigate what causality has been embedded in the model instead of running simulations and calculating probabilities. In particular, if the models used are climate models as in [92], their inability to effectively describe (perform in "prime time") the real-world processes [50,95–100] makes the entire endeavour futile. Another notion these authors use is *exogeneity*, which is related to the so-called *causal graph*, reflecting the assumed dependencies among the studied variables. Specifically, they state "a sufficient condition for *X* to be exogenous wrt any variable is to be a top node of a causal graph".

Here, we will use the simple example of Section 4.2, temperature–clothes weight–sweat, to show that using the quantities FAR (or PN) and PS may give spurious results that do not correspond to necessary or sufficient conditions for causality, at least with their meaning in our paper.

We use the two-valued random variables $\underline{x}, \underline{y}, \underline{z}$ to model the states of temperature, clothes weight, and sweat, respectively. We designate the following states:

$x = 1$: being hot *above* a threshold;
$y = 1$: wearing clothes with weight *below* a threshold;
$z = 1$: sweat quantity *above* a threshold;

and the opposite states with $x = 0, y = 0, z = 0$, respectively. We choose the threshold of temperature so that $P\{\underline{x} = 0\} = P\{\underline{x} = 1\} = 0.5$ and that of clothes weight so that $P\{\underline{y} = 0\} = P\{\underline{y} = 1\} = 0.5$. We choose a small probability, 0.05, of wearing light clothes when cold, or heavy clothes when hot, i.e., $P\{\underline{y} = 1 | \underline{x} = 0\} = P\{\underline{y} = 0 | \underline{x} = 1\} = 0.05$ (generally, we avoid choosing zero probabilities; rather the minimum value we choose is 0.05).

Using the definition of conditional probability,

$$P\{\underline{y} = y | \underline{x} = x\} = \frac{P\{\underline{y} = y, \underline{x} = x\}}{P\{\underline{x} = x\}}, \tag{A12}$$

we find the probability matrix $A$ with elements $a_{ij} = P\{\underline{x} = i, \underline{y} = j\}$ as follows:

$$A = \begin{bmatrix} 0.475 & 0.025 \\ 0.025 & 0.475 \end{bmatrix} \begin{matrix} x = 0 \\ x = 1 \end{matrix} \quad . \tag{A13}$$
$$\begin{matrix} y = 0 & y = 1 \end{matrix}$$

Now, we assign plausible values to the conditional probabilities of high sweat, $P\{\underline{z} = 1 | \underline{x} = x, \underline{y} = y\}$, as follows:

Cold, heavy clothes: $\qquad P\{\underline{z} = 1 | \underline{x} = 0, \underline{y} = 0\} = 0.2$

Cold, light clothes: $\qquad P\{\underline{z} = 1 | \underline{x} = 0, \underline{y} = 1\} = 0.1$

Hot, heavy clothes: $\qquad P\{\underline{z} = 1 | \underline{x} = 1, \underline{y} = 0\} = 0.95$

Hot, light clothes: $\qquad P\{\underline{z} = 1 | \underline{x} = 1, \underline{y} = 1\} = 0.80$

Again, we have avoided setting any of the conditional probabilities to 0 (or 1), and we have used multiples of 0.05 for all of them.

Using the definition of conditional probability in the form

$$P\{\underline{z} = z | \underline{x} = x, \underline{y} = y\} = \frac{P\{\underline{z} = z, \underline{y} = y, \underline{x} = x\}}{P\{\underline{y} = y, \underline{x} = x\}},$$

(A14)

we find the joint probabilities for each of the triplets $\{x, y, z\}$ that are shown in Table A1.

**Table A1.** Joint probabilities $P\{\underline{x} = x, \underline{y} = y, \underline{z} = z\}$ for all triplets $\{x, y, z\}$

| $x$ | $y$ | $z = 0$ | $z = 1$ |
|---|---|---|---|
| 0 | 0 | 0.38 | 0.095 |
| 0 | 1 | 0.0225 | 0.0025 |
| 1 | 0 | 0.00125 | 0.02375 |
| 1 | 1 | 0.095 | 0.38 |
| | $P\{\underline{z} = z\} =$ | 0.49875 | 0.50125 |

Now, assume that we let an "artificial intelligence entity" (AIE) decide on causality based on the probability rules of the Hannart et al. [92] framework. Our AIE has access to numerous videos of people and is "trained" to assign accurate values of $y$ and $z$, referring to clothes and sweat, based on the images in videos. In the video images, no thermometers are shown and, thus, our AIE cannot assign values of $x$, nor can it be aware of the notion of temperature. Our AIE tries to construct a causal graph putting, say, $\underline{y}$ as a top node and $\underline{z}$ as an end node; hence, it assumes that $\underline{y}$ is exogenous. Based on the huge information it can access, our AIE can (a) claim that it has constructed a prediction model based on one part of the data (e.g., using the so-called deep-learning technique) and, hence, is able to perform "in silico experimentation" (even though this is not absolutely necessary) and (b) accurately estimate the joint and conditional probabilities related to $\{y, z\}$ using either the model, the data, or both. Provided that the dataset is large enough, it will come up with the true values for the conditional probabilities, which are $b_{ij} = P\{\underline{y} = i, \underline{z} = j\}$ and $c_{ij} = P\{\underline{z} = j | \underline{y} = i\}$, and form the matrices $B$ and $C$, respectively, with values as follows:

$$B = \begin{bmatrix} 0.38125 & 0.11875 \\ 0.1175 & 0.3825 \end{bmatrix} \begin{matrix} y = 0 \\ y = 1 \end{matrix} \quad , \quad C = \begin{bmatrix} 0.7625 & 0.2375 \\ 0.235 & 0.765 \end{bmatrix} \begin{matrix} y = 0 \\ y = 1 \end{matrix} .$$
$$\phantom{B = } z = 0 \quad z = 1 \phantom{\quad , \quad C = } z = 0 \quad z = 1$$

(A15)

Here, the true values $b_{ij}$ have been determined from the values of Table A1 noting that

$$b_{ij} = P\{\underline{y} = i, \underline{z} = j\} = P\{\underline{z} = j, \underline{y} = i, \underline{x} = 0\} + P\{\underline{z} = j, \underline{y} = i, \underline{x} = 1\}$$

(A16)

and the true values $c_{ij}$ have been determined from the definition of conditional probability:

$$P\{\underline{z} = z | \underline{y} = y\} = \frac{P\{\underline{z} = z, \underline{y} = y\}}{P\{\underline{y} = y\}}.$$

(A17)

Our AIE will then implement the causality conditions of sweat on clothes weight, assigning $p_0 = P\{\underline{z} = 1 | \underline{y} = 0\} = 0.2375$ and $p_1 = P\{\underline{z} = 1 | \underline{y} = 1\} = 0.765$. It will further calculate the probability of necessary causality as PN = 0.690, and the probability of sufficient causality even higher, PS = 0.692. Hence, our AIE will inform us that there is all necessary and sufficient evidence that light clothes cause high sweat.

Now, coming to the study by Verbitsky et al. [93], we notice that it assumes that "each time series is a variable produced by its hypothetical low dimensional system of dynamical equations" and uses the technique of distances of multivariate vectors for reconstructing the system dynamics. As demonstrated in Koutsoyiannis [101], such assumptions and techniques are good for simple toy models but, when real-world systems are examined, low dimensionality appears as a statistical artifact because the reconstruction actually needs an incredibly high number of observations to work, which are hardly available. The fact that the sums of multivariate vectors of distances is a statistical estimator with huge uncertainty is often missed in studies of this type, which treat data as deterministic quantities to obtain unreliable results. We do not believe that the Earth system and Earth processes (including global temperature and $CO_2$) are of low dimensionality, and we deem it unnecessary to discuss the issue further. We only note the fact that global temperature and $CO_2$ virtually behave as Gaussian, which enables reliable estimation of standard correlations and dismiss the need to use the overly complex and uncertain correlation sums.

*Appendix A.5. Additional Graphical Depictions*

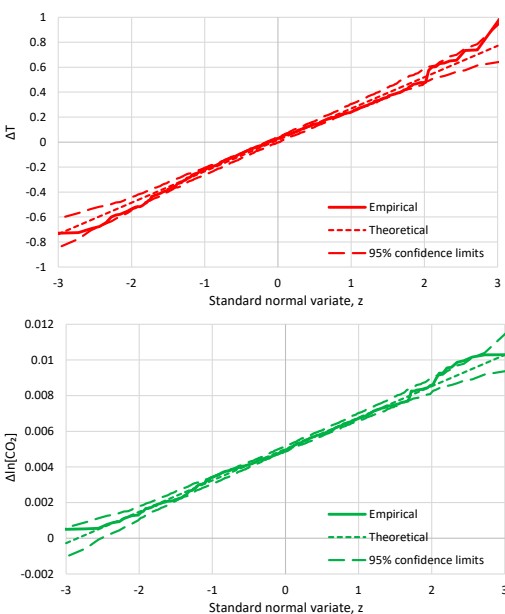

**Figure A1.** Normal probability plots of $\Delta T$ and $\Delta\ln[CO_2]$ where $T$ is the UAH temperature and $[CO_2]$ is the $CO_2$ concentration at Mauna Loa at monthly scale.

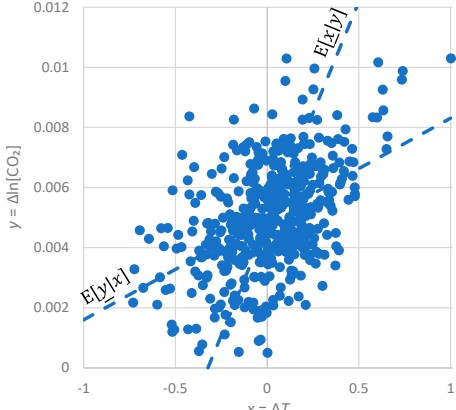

**Figure A2.** Scatter plot of $\Delta T$ and $\Delta\ln[CO_2]$ where $T$ is the UAH temperature and $[CO_2]$ is the $CO_2$ concentration at Mauna Loa at monthly scale; the two quantities are lagged in time using the optimal lag of 5 months (Table 1). The two linear regression lines are also shown in the figure.

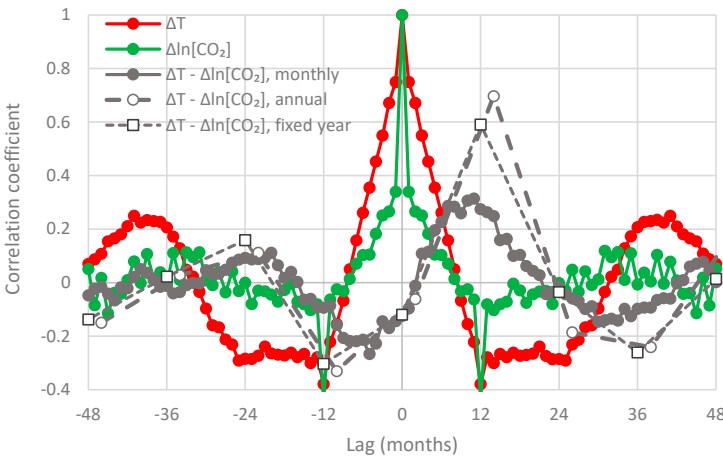

**Figure A3.** Auto- and cross-correlograms of the differenced time series of UAH temperature and Barrow $CO_2$ concentration.

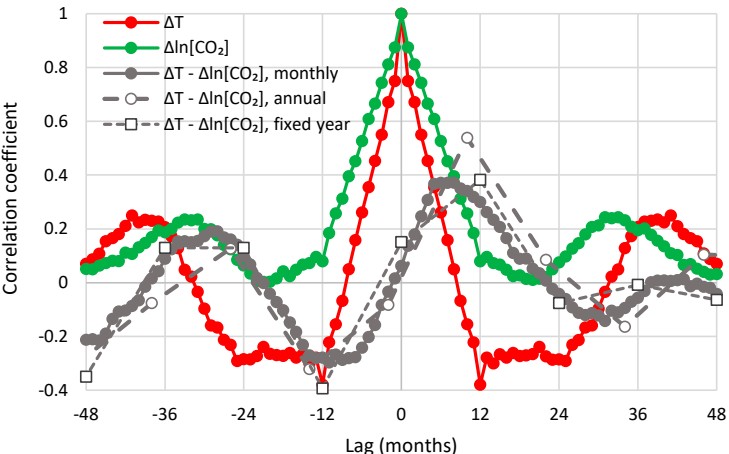

**Figure A4.** Auto- and cross-correlograms of the differenced time series of UAH temperature and South Pole $CO_2$ concentration.

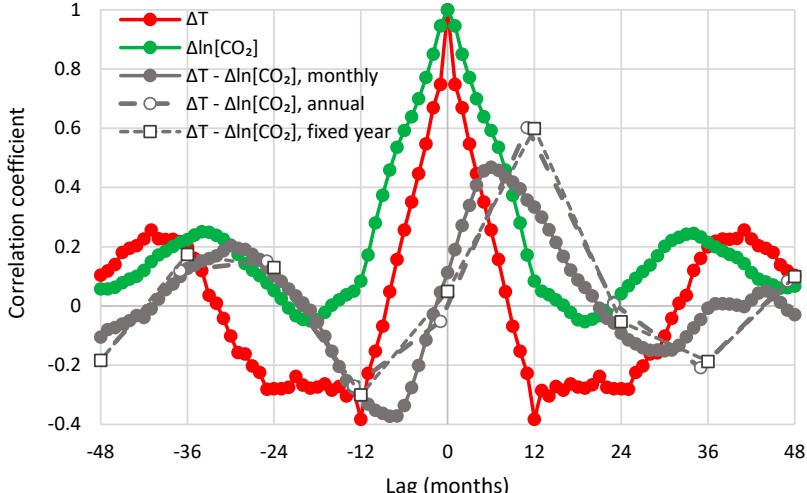

**Figure A5.** Auto- and cross-correlograms of the differenced time series of UAH temperature and global $CO_2$ concentration.

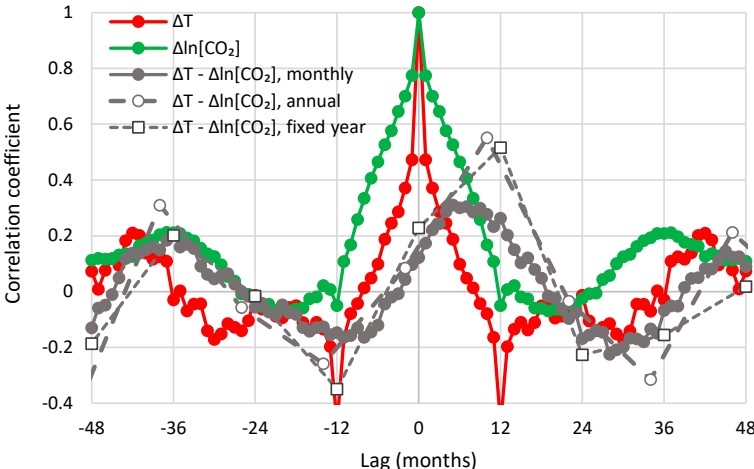

**Figure A6.** Auto- and cross-correlograms of the differenced time series of CRUTEM4 temperature and Mauna Loa $CO_2$ concentration.

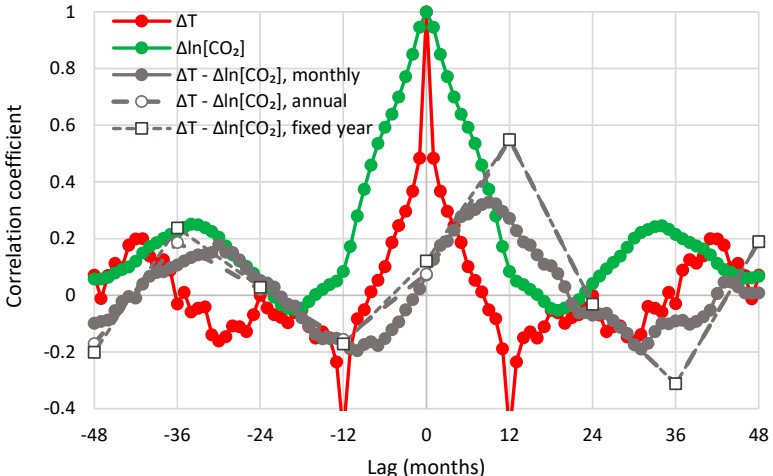

**Figure A7.** Auto- and cross-correlograms of the differenced time series of CRUTEM4 temperature and global $CO_2$ concentration.

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
