# Peer review of "Atmospheric Temperature and CO2: Hen-Or-Egg Causality?"

_sci, doi:10.3390/sci2040083_

Round 1
Reviewer 1 Report
# The authors proposed a new approach to analyze the well-known problem. Similar studies are generally performed on longer time series (https://doi.org/10.1038/srep21691, https://doi.org/10.1007/s10584-013-1007-x ). It would be interesting to see how the results based on the proposed approach in this paper compare with the previous works.
# The literature review misses recent/relevant works. For instance,
https://doi.org/10.5194/gmd-12-4053-2019
https://doi.org/10.1175/BAMS-D-14-00034.1
Author Response
(as it appears in https://www.mdpi.com/2413-4155/2/3/72)Reply to Reviewer 1 – Yog Aryal
We thank Yog Aryal for his review [1], his positive evaluation of Version 1 of our paper [2] and his suggestions of relevant references [3-6].
In fact, the references [3-4] had already been reviewed extensively in our Version 1. It is true, however, that these refer to time spans longer than in our study. In our Version 2, and specifically in the new Appendix 1, we explain our reasoning why we delimit our study period to 1980-2019 and do not use controversial data of earlier periods.
Also, in Version 2 we have included the Reviewer’s suggestions [5-6] within the new Appendix 4, where we explain our theoretical disagreements with the methodologies in both these papers. Because of these disagreements, we do not proceed in comparisons of results.
References
1. Aryal, Y. Review of Atmospheric Temperature and CO₂: Hen-or-Egg Causality? (Version 1). Sci 2020, https://www.mdpi.com/2413-4155/2/3/72 (posted and accessed on 25 September 2020).
2. Koutsoyiannis, D.; Kundzewicz, Z.W. Atmospheric Temperature and CO2: Hen-or-Egg Causality? (Version 1). Sci 2020, 2, 72.
3. Stips, A.; Macias, D.; Coughlan, C.; Garcia-Gorriz, E.; Liang, X.S. On the causal structure between CO2 and global temperature. Sci. Rep. 2016, 6, 21691, doi:10.1038/srep21691.
4. Stern, D.I.; Kaufmann, R.K. Anthropogenic and natural causes of climate change. Clim. Chang. 2014, 122, 257–269.
5. Verbitsky, M.Y., Mann, M.E., Steinman, B.A., and Volobuev, D.M.: Detecting causality signal in instrumental measurements and climate model simulations: global warming case study. Geosci. Model Dev. 2019, 12, 4053–4060, doi: 10.5194/gmd-12-4053-2019.
6. Hannart, A.; Pearl, J.; Otto, F.E.L.; Naveau, P.; Ghil, M. Causal counterfactual theory for the attribution of weather and climate-related events. Bulletin of the American Meteorological Society 2016, 97 (1), 99-110.
Reviewer 2 Report
Review of “Atmospheric temperature and CO2: Hen-or-egg causality?”
Summary of manuscript and general comments
It is a well-known observation in statistics that, “correlation does not imply causation”. A large body of literature exists arguing that global temperature (I will use “T” for short) trends are somewhat correlated to atmospheric CO2 trends. Various theoretical and/or computer model-based analyses have argued that changes in atmospheric CO2 concentrations are a major driver (if not THE primary driver) of global temperature trends, i.e., that the direction of causation is CO2 → global temperatures. However, in their manuscript, the authors (Koutsoyiannis and Kundzewicz) essentially argue that from an empirical point of view, the net direction of causality of the CO2/T correlation has not yet been satisfactorily established. Moreover, they present intriguing empirical evidence that suggests the net direction of causation might be the opposite, or at least that there are feedbacks in both directions.
The theory that (a) atmospheric CO2 is the primary driver of global temperature trends and (b) that human activity is the only significant driver of atmospheric CO2 trends since the start of the Industrial Revolution currently dominates the relevant scientific literature. In particular, the theory was the primary motivation for the setting up in the 1980s of the Intergovernmental Panel on Climate Change (IPCC), whose reports have been highly influential and are often assumed to represent the “scientific consensus” on all matters related to “climate change”. And, the theory underpins much of the framing of the discussion within the IPCC reports.
Moreover, the theory has become highly politicised and even daring to ask questions about it has become “politically incorrect”.
However, as scientists, I believe we should be less concerned about the political correctness of a scientific theory and be more interested in its scientific robustness. Indeed, Sarewitz (2011) also notes that encouraging active scientific inquiry into “consensus statements” is ultimately also important for policymakers:
“The very idea that science best expresses its authority through consensus statements is at odds with a vibrant scientific enterprise. Consensus is for textbooks; real science depends for its progress on continual challenges to the current state of always-imperfect knowledge. Science would provide better value to politics if it articulated the broadest set of plausible interpretations, options and perspectives, imagined by the best experts, rather than forcing convergence to an allegedly unified voice” – Sarewitz (2011)
Therefore, I agree with the authors that we should be encouraging open-minded scientific inquiry into our prevailing paradigms. If our current paradigms are truly robust, then they should easily survive critical scientific inquiry, and so we should have nothing to fear from such inquiry. On the other hand, if our paradigms are unable to withstand such inquiry, this would indicate that we may be in need of a better paradigm (Kuhn, 1962).
The authors conclude on the basis of their manuscript,
“This poses challenging scientific questions of interpretation and modelling for further studies. In our opinion, scientists of the 21st century should have been familiar with unanswered scientific questions, as well as with the idea that complex systems resist simplistic explanations”
In light of my comments above, even the epistemological value alone of this manuscript of this conclusion (and the arguments leading to it) make the paper worthy of consideration for publication. The intriguing results implied by Figures 1 and 2, the timeliness of the analysis following several months of the many COVID-19 national lockdowns (with a significant reduction in anthropogenic CO2 emissions – as opposed to concentrations) also merit publication. Finally, the analysis itself highlights key conflicts between the simple narrative of the theoretical understanding of the prevailing paradigms on CO2 and empirical observations. Even if these conflicts can ultimately be resolved within the current paradigms, the authors findings demonstrate that the inter-relationships between CO2 and T are more complex than typically assumed. Therefore, the authors’ findings should be of interest to the scientific community.
However, as it stands, there is some room for improvement. Below I will make a number of comments and suggestions for how I think the manuscript could be improved. Although some of these are just suggestions, I recommend the authors consider these comments and revise their manuscript accordingly.
In my opinion, my recommendations could be implemented with relatively minor revisions, but if they prompt the authors to make more substantive revisions, this is acceptable to me. Overall, I think the manuscript is an important one, but needs some improvement first.
Recommendation: Minor revisions necessary
Specific comments
1. Acknowledge the wider epistemological framework for your framing of the topic
I first of all recommend reading our recent paper, Connolly et al. (2020). It is a long paper, but I think you would find Section 3.1 particularly relevant.
Note our suggestion that there are currently 4 distinct paradigms on the relationship between T & CO2. This distinction between four paradigms overlaps somewhat with your two proposed directions of causality, however it suggests that there are additional complexities that need to be considered other than the net direction of causality. Using your schematic in the graphical abstract, Paradigms 1 and 2 would imply egg to hen, while Paradigms 3 and 4 would imply hen to egg. Paradigms 2 and 3 suggest that at least some feedback occurs between egg and hen.
I leave it up to you whether you decide to incorporate into your framework this concept of there being at least four paradigms, rather than just two. But, you should at least refer to it.
However, a more serious limitation of your current framing is that as far as I see it the hen/egg correlation is not strictly T→CO2 vs. CO2→T. Rather, as I see it the two sides of your proposed hen/egg relationship are:
- “T→CO2”. Changes in global temperatures alter the balances between several of the natural sinks and sources of atmospheric CO2. The net effect of these leads to an increase in atmospheric CO2 with global warming (and equivalently a decrease in atmospheric CO2 with global cooling)
- “CO2→T”. I argue that this proposed causality can actually be split into two components.
- Anthropogenic CO2 emissions are the sole reason for the observed multi-decadal increase in atmospheric CO2 since 1958 (and if we assume the Antarctic ice core estimates are reliable, since the start of the Industrial Revolution)
- Changes in atmospheric concentrations of CO2 have been the primary driver of global temperature changes over this period
(2a) and (2b) are actually separate hypotheses which should be evaluated separately. They also are both compatible with (1). This has several implications:
For example, because (1) does not strictly depend on what the actual drivers of global temperature changes are, it could apply regardless of whether global warming is mostly (or entirely) natural or mostly (or entirely) anthropogenic. In the latter case, this could therefore act as a feedback to the 2(b) causality, while in the former case it could be independent of (2).
Because 2(a) and 2(b) are separate hypotheses, it is possible that one of them might be valid, but the other not. If 2(b) transpires to be invalid, but 2(a) transpires to be valid, then CO2 might be increasing due to anthropogenic emissions, without being the primary driver of global temperatures. In that case, it might be that neither the “hen” or “egg” causality would apply. One might then ask why trends in CO2 and T appear to have been correlated in the modern era. Actually, as I will discuss below, depending on the datasets you consider, the apparent correlations are not always as compelling as is commonly claimed.
I suggest that, from an epistemological perspective (which I think your paper is trying to present), we should systematically consider the various combinations of (1), (2a) and (2b), rather than simply treating this as a 1 vs 2 “hen vs egg” scenario. That would give us at least 2^3 = 8 scenarios rather than just two:
- 1 = true; 2a = true; 2b = true
- 1 = true; 2a = true; 2b = false
- 1 = true; 2a = false; 2b = false (i.e., exclusively “T→CO2”)
- 1 = true; 2a = false; 2b = true
- 1 = false; 2a = true; 2b = true (i.e., exclusively “CO2→T”)
- 1 = false; 2a= true; 2b = false
- 1 = false; 2a = false; 2b = true
- 1 = false; 2a = false; 2b = false
I would also recommend considering the possibility that each of these three proposed causalities might be “partially true”, rather than either true or false. That would give us at least 3^3 = 27 scenarios rather than the 8 scenarios outlines above.
With regards to how much of the recent warming is natural vs. anthropogenic, I suggest considering Sections 4 and 5 of the Connolly et al. (2020) paper I mentioned above, where we show that there is still considerable ongoing debate over this challenging question in the literature.
2. How strong is the correlation between T and CO2?
In Section 2, you indicate that the correlations between T and CO2 might be different on different timescales. You note that part of this may be related to uncertainties over proxy reconstructions. Nonetheless, you suggest that there may be a reasonable correlation over the Phanerozoic era, by comparing one T reconstruction and 3 CO2 reconstructions in Figure 3. It might also be worth referring to the debates between Shaviv and Veizer (2003) who argued on the basis of other reconstructions that T and CO2 are not well correlated over this period. But, then noting that this claim was disputed by Rahmstorf et al. (2004) and Royer et al. (2005). Both of these latter two papers generated some correspondence (see references at the end).
You note, on the basis of Antarctic ice core estimates, that there appears to be a stronger correlation between T and CO2 during the glacial/interglacial transitions of the last 400,000 years, but that there is some evidence that the direction of causation may be T→CO2 on these timescales.
The rest of your analysis largely focuses on the modern instrumental records, i.e., after the Mauna Loa record began in 1958. This is reasonable, and I think it is good that you note the distinction between correlations on short timescales vs longer timescales.
However, have you considered the debates over the intermediate timescales? I would recommend consider the literature cited in Section 3.1 of Connolly et al. (2020) which I mentioned above. In particular, I would draw your attention to the debates over whether the stomatal based estimates of paleo-CO2 over the last few millennia are more or less reliable than the Antarctic ice core estimates. These stomatal-based estimates imply that the pre-industrial variability of atmospheric CO2 concentrations was significantly greater than that implied by the Antarctic ice cores.
You may also be interested in the debates over the pre-1958 instrumental measurements prompted by Beck’s 2007 and 2008 papers (Beck sadly passed away shortly afterwards).
3. Improve the discussion of the relevant literature
Over the years, several researchers have presented similar arguments to you, from various perspectives. Typically, these studies are vehemently disputed by other researchers. In turn, these rebuttals are typically countered by the original authors. Perhaps some of the vitriol in such correspondence is related to the politically-charged nature of the topic (as mentioned in my preamble).
At any rate, I think it is important that you acknowledge more of this earlier literature making similar arguments, and also address the criticisms of that earlier literature where relevant to their analysis.
For some of the key references, I would again recommend you consider the literature cited in Section 3.1 of Connolly et al. (2020) which I mentioned above. However, I would particularly draw your attention to the following:
- Ole Humlum et al. (2013)’s paper which somewhat overlaps with your analysis. Link here: https://doi.org/10.1016/j.gloplacha.2012.08.008 This study was critiqued by Richardson (2013): https://doi.org/10.1016/j.gloplacha.2013.03.011 and Masters & Benestad (2013): https://doi.org/10.1016/j.gloplacha.2013.03.010
- Murry Salby has also been making similar arguments. As far as I know, he hasn’t published any of this analysis in a peer-reviewed journal yet. But, he has given a few talks. Here’s one of his more recent ones (2018): https://www.youtube.com/watch?v=rohF6K2avtY
4. On your proposed mechanism for T→CO2
In Section 6, you briefly postulate some of the mechanisms by which increasing T could cause increasing CO2.
I agree with you that increasing global temperatures should lead to increasing atmospheric CO2 from increased soil respiration. Indeed, in a recent paper - ÓhAiseadha et al. (2020), we briefly pointed out that this leads to the surprising fact that the net night-time soil warming caused by wind farms is probably leading to an increase in biological CO2 emissions which may well be counteracting some (or all) of the reduction in anthropogenic CO2 emissions the wind farms are hoped to cause.
You might find some of the references we cite in Section 4.2.4 of ÓhAiseadha et al. (2020) relevant for your arguments.
However, I would suggest that there are other mechanisms by which global temperatures could alter atmospheric CO2 concentrations – and broadly they typically are of the same sign, i.e., more warming leading to increased atmospheric CO2 concentrations.
For instance, I would also note that the solubility of CO2 in water decreases with increasing temperature. So, it is plausible that increasing temperature could also cause increasing CO2 through outgassing from the oceans. This is something we are considering for a manuscript that we are working on. However, we are still evaluating this hypothesis, as we are realising there are several unresolved issues associated with the transfer of CO2 across the surface ocean/air boundary.
Nonetheless, the transfer of CO2 back and forth between the surface oceans and the atmosphere is an important component of the annual CO2 fluxes. Therefore, changes in average SST may also be contributing to changes in atmospheric CO2.
Also, we still haven’t published this formally, but our current best explanation for the high seasonal variability of the Barrow (and other Arctic CO2 monitoring stations, e.g., Alert, Canada) is that it is probably related to the seasonality of sea ice. That is, when the oceans are covered by sea ice, no gaseous exchange can occur, but once the sea ice melts in the Arctic summer, CO2 can be exchanged. And because the sea surface temperature is relatively cold, the CO2 solubility is relatively high, i.e., CO2 enters the oceans.
If this hypothesis is correct, then it suggests that changes in average sea ice cover from changing global temperatures might also alter atmospheric CO2.
These proposed mechanisms are less well grounded in the literature than the soil respiration mechanism. However, I suggest that you should at least consider the possibility of other mechanisms.
5. Minor comments
1. At the end of the Introduction, you state, “Interestingly, Figure 2 also shows a rapid growth in emissions after the 2008-2009 global financial crisis…”. However, Figure 2 only seems to plot the results for 2017-2020. Am I missing something?
2. In Section 2, you suggest that the key findings of John Tyndall (1865) were preceded by an earlier study by Eunice Foote (1856). I have seen this claim being made in some circles, however it doesn’t stand up to scrutiny. I suspect the popularity of the claim seems to be more related to the current fashionability of judging the importance of an individual researcher’s work based on their gender, race or nationality, rather than an actual genuine interest in establishing accurate history. That is, within certain circles the fact that Eunice Foote was an American woman while John Tyndall was an Irish man (as am I, for what it’s worth) is apparently of more importance than their actual findings. However, I suggest you read Foote (1856) and Tyndall (1865) more closely.
Tyndall was effectively looking at the infrared activity of the various (then known) atmospheric gases, and identified that H2O and CO2 (as well as several other trace gases, including CH4) were infrared-active, while the bulk gases, N2 and O2 were not (Argon had not been discovered at that stage, but is also IR-inactive). Building on Joseph Fourier’s work (a French man, for what it’s worth) in the 1820s, Tyndall suggested that this could indicate that changes in atmospheric H2O might be involved in the glacial/interglacial transitions (which as you note currently are believed to be driven by Milankovitch orbital cycles). Arrhenius (1896) later argued that variations in CO2 were a better candidate. This was in turn later disputed by Angstrom (1901) and Simpson (1929), for example.
Later, Callendar (1938) revisited Arrhenius’s theory, and argued that anthropogenic CO2 emissions since the 19th century could also explain the observed Northern Hemisphere warming from the end of the 19th century up to 1930s. It is worth noting that Simpson was one of the commenters on that paper and was somewhat sceptical. It is also worth noting that the warming Callendar identified was later followed by Northern Hemisphere cooling from the 1950s to 1970s. However, during this Northern Hemisphere cooling period, several researchers began developing theoretical frameworks and computer models that argued CO2 was the primary driver of global temperatures via altering the rates of infrared cooling of the Earth, e.g., Plass (1959); Manabe & Wetherald (1975). The current paradigm that CO2 is the primary driver of global temperatures builds upon these models by Plass, Manabe & Wetherald et al.
However, Foote’s study was not looking at infrared absorption. She was looking at the absorption of incoming sunlight. Also, she failed to replicate Tyndall’s identification of H2O as being the dominant infrared absorber in the Earth’s atmosphere. Moreover, in my opinion, she didn't present any great insights to explain her findings. Compare her 1.5 page long article to the ~40 page treatise of Tyndall which presented rigorous documentation of assumptions, tests and preliminary measurements as well as detailed discussions of the context of his findings for the then-knowledge of chemistry (as well as his comments on glacial cycles, which was actually a relatively minor aspect of the study).
If Foote’s study of the absorption of incoming sunlight is supposed to represent a description of the “greenhouse effect” theory, then it begs the question of what exactly is the “greenhouse effect theory” she discovered – it certainly isn’t the one that current climate models use (which is based on theoretical calculations based on infrared fluxes within the atmosphere). And, if it was merely the fact that she was considering the absorption of radiation by the atmosphere (neglecting the emission of radiation by the atmosphere), then Fourier’s 1920s research predates hers.
3. Page 5. A typo: “While we occasionally use the Ganger [sic] statistical test…”
References cited in this review
Angstrom, K. 1901. KNUT ANGSTROM ON ATMOSPHERIC ABSORPTION. Mon. Wea. Rev. (1901) 29 (6): 268. https://doi.org/10.1175/1520-0493(1901)29[268a:KAOAA]2.0.CO;2
Callendar, G.S., 1938. The artificial production of carbon dioxide and its influence on temperature. Quat. J. Roy. Met. Soc. 64:223-240. https://doi.org/10.1002/qj.49706427503
Connolly et al., 2020. “How Much Human-Caused Global Warming Should We Expect with Business-As-Usual (BAU) Climate Policies? A Semi-Empirical Assessment” Energies. 13:1365. https://doi.org/10.3390/en13061365
ÓhAiseadha, C. et al., 2020. Energy and climate policy – an evaluation of global climate change expenditure 2011-2018. Energies, 13:4839. https://doi.org/10.3390/en13184839
Kuhn T.S. 1962. The structure of scientific revolutions. University of Chicago Press. 264pp.
Manabe, S.; Wetherald, R.T. (1975). The Effects of Doubling the CO2 Concentration on the climate of a General Circulation Model. J. Atmos. Sci. 32: 3–15. https://doi.org/10.1175/1520-0469(1975)032<0003:TEODTC>2.0.CO;2
Plass, G.N. (1959). Carbon Dioxide and Climate. Sci. Am. 201:41-47. https://www.jstor.org/stable/24940327
Sarewitz D 2011 The voice of science: let’s agree to disagree Nature 478 7–7
Simpson, G.C. 1929. Past Climates. Nature 124, 988–991 (1929). https://doi.org/10.1038/124988a0
Shaviv, N.J. and Veizer, J., 2003. Celestial driver of Phanerozoic climate? Geol. Soc. Am. Today. https://www.geosociety.org/gsatoday/archive/13/7/pdf/i1052-5173-13-7-4.pdf
Rahmstorf, S. et al. 2004. Cosmic rays, carbon dioxide, and climate. Eos, 85:38-41. https://doi.org/10.1029/2004EO040002
Comment and reply to Rahmstorf et al. 2004. https://doi.org/10.1029/2004EO480004 and https://doi.org/10.1029/2004EO480005
Royer, D.L. et al., 2005. CO2 as a primary driver of Phanerozoic climate. Geol. Soc. Am. Today. https://www.geosociety.org/gsatoday/archive/14/3/pdf/i1052-5173-14-3-4.pdf
Comment and reply on Royer et al. 2005. https://www.geosociety.org/gsatoday/archive/14/7/pdf/i1052-5173-14-7-18b.pdf
Shaviv, N.J., 2005. On climate response to changes in the cosmic ray flux and radiative budget. J. Geophys. Res. https://doi.org/10.1029/2004JA010866
Author Response
(as it appears in https://www.mdpi.com/2413-4155/2/3/72)Reply to Reviewer 2 – Ronan Connolly
We thank Ronan Connolly for his very detailed review [1], his positive evaluation of Version 1 of our paper [2] and his suggestions of a great deal of relevant references, most of which we were unaware of.
In Version 2 we reviewed and cited virtually all of the references he suggested. Exceptions are a suggestion for a videotaped talk and some references on general epistemological issues (e.g. Kuhn on the structure of scientific revolutions). Instead of citing the latter, we thought it fairer to recognize the review by Connolly, which is available in open access, as an individual literature entry and cite it per se. Indeed, due to its extent, detail, broad scope and original ideas, we believe that this review is a citable item (see our Conclusions section).
Section “How strong is the correlation between T and CO2?”: In section 2 of Version 2 we have included the discussion about the study by Shaviv and Veizer [3] and all related commentaries. We were reluctant to include analyses for intermediate time scales, as we have not studied them, but we cited Connolly et al. [4] in our Conclusions Section. About Beck’s [5] paper, we have devoted the new Appendix 1 (see also our replies to the other reviewers). Section “Improve the discussion of the relevant literature”. In addition to what we have already noted above, we are thankful to the Reviewer for pointing out the study by Humlum et al. [5] and its commentaries, all of which we have cited in Version 2. Section “On your proposed mechanism for T→CO2”: In our Section 6 we have cited ÓhAiseadha et al. [7] but we do not refer to the effect of wind farms as we deemed it not relevant to our study. We are thankful for the idea about the effect of temperature on solubility of CO₂ in water and we have included it as an additional possible physical explanation. However, we believe that the quantification of that effect is not easy and needs tricky calculations, which are out of the scope of our paper and perhaps of our own expertise. We thus look forward to Connolly’s publication reporting his results on this. Section “Minor comments”: We are thankful for the careful reading and the spotting of two typos. As per the discussion about Foote [8], in view of the Connolly’s [1] analysis we have changed our phraseology in the sentence where we refer to Foote [8], but we did not expand our discussion, preferring to refer to Connolly [1] and another couple of references for the details. 1. Connolly, R. Review of “Atmospheric temperature and CO2: Hen-or-egg causality?” (Version 1). Sci 2020, https://www.mdpi.com/2413-4155/2/3/72 (posted and accessed on 09 October 2020) 2. Koutsoyiannis, D.; Kundzewicz, Z.W. Atmospheric Temperature and CO2: Hen-or-Egg Causality? (Version 1). Sci 2020, 2, 72. 3. Shaviv, N.J.; Veizer, J. Celestial driver of Phanerozoic climate?. Geol. Soc. Am. Today 2003, 13 (7), 4-10. 4. Connolly, R.; Connolly, M.; Carter, R.M.; Soon, W. How Much Human-Caused Global Warming Should We Expect with Business-As-Usual (BAU) Climate Policies? A Semi-Empirical Assessment. Energies 2020, 13, 1365. 5. Humlum, O.; Stordahl, K.; Solheim, J.E. The phase relation between atmospheric carbon dioxide and global temperature, Global and Planetary Change 2013, 100, 51-69, doi: 10.1016/j.gloplacha.2012.08.008. 6. Beck, E.-G. 180 Years of atmospheric CO2 gas analysis by chemical methods. Energy and Environment 2007, 18 (2), 259-282. 7. ÓhAiseadha, C.; Quinn, G.; Connolly, R.; Connolly, M.; Soon, W. Energy and Climate Policy—An Evaluation of Global Climate Change Expenditure 2011–2018. Energies 2020, 13, 4839. 8. Foote, E. Circumstances affecting the heat of the sun’s rays. Am. J. Sci. Arts 1856, 22, 382–383. Reviewer 3 Report Review by Stavros Alexandris on sci-02-00072-v2 document. I think that the submitted work (sci-02-00072-v2) is a robust and comprehensive document, carefully prepared. The main findings (results) are consistent with the research made and are reflected in the conclusions. The authors use reliable sources of information and have an excellent and extended discussion in all almost sections of the document. Indeed, the smart title of the work chosen by the authors with the crucial question "Hen-Or-Egg Causality?" gives an essential dimension of the problem since a substantiated answer to the query would be very interesting to know not only to the scientific community but the global society. A clear answer - regardless of its direction (T→CO2 or T←CO2) - would then answer the question: Global warming has natural or anthropogenic causes and, what is the magnitude of this direction? Thus, the title of the manuscript wording entirely reflects the content of the paper. The authors examine both causality directions thoroughly and develop a stochastic representation, trying to explain and detect through basic and essential elements the conditions of causality. I would say that the authors in this section have indeed been impartial, and they achieve convincing results. At a first glance in Figures 1 & 2, with a joined-up approach, one could discover the self-evident and logical argument on the relation between CO2 concentration and global energy emissions of CO2 in the last four years. This argument is very clearly analyzed and commented on by the authors. The interesting points of this paper are the methodologies and statistical approaches that the authors have developed to ensure the presentation of results in an efficient way. An original idea of the authors is the graphical representation of the results, offering better visualization of the lag direction, and magnitude in all ranges of the time series. Besides all the above, I would like to point out something that gives substantial value to this work, even though that might not be visible to readers. The two authors of this work are unquestionably recognizable scientists from the international scientific community, not only from their writing and research works but also their different views and positions on the controversial topic of climate change. This collaboration is a model of scientific partnership that could teach a lot the new researchers and the academic community. The real consensus lies in documenting findings beyond any prejudice. I would suggest as an improvement of the work a more extensive introduction to include important bibliographic references to the historical background that strengthen the main finding of the paper. e.g. Beck (2007) [1] https://journals.sagepub.com/doi/abs/10.1177/0958305X0701800206 Also, I would suggest an extension of this section 6. (Physical interpretation) [1] Beck, E. G. (2007). 180 years of atmospheric CO2 gas analysis by chemical methods. Energy & Environment, 18(2), 259-282. Author Response We thank Stavros Alexandris for his review [1], his positive evaluation and approval of Version 1 of our paper [2], and his suggestions of a relevant reference [3]. We are very pleased that the Reviewer noted strengths of our paper, its novelty and impartiality, and that he found our results convincing. He also noted our “different views and positions on the controversial topic of climate change.” We gladly read his words: “This collaboration is a model of scientific partnership that could teach a lot the new researchers and the academic community. The real consensus lies in documenting findings beyond any prejudice.” Indeed, having various views on a range of issues, we were united in the curiosity of listening to what the data say, without making subjective assumptions. The reviewer suggests that we extend our Introduction and strengthen the section on Physical interpretation. We have reacted to this suggestion by adding new material. As regards the Introduction we note that we wanted to keep it brief and focused on our motivation, as well as the logic and structure of our paper. Therefore, the new material has been added in the next section 2 (“Temperature and Carbon Dioxide— From Arrhenius and Palaeo-Proxies to Instrumental Data”), which contains most of the literature review. Note that Version 2 includes a lot of new references (a total increase of 66% with respect to Version 1). The reviewer also suggests citing Beck [3], which we did in the new Appendix 1. We also cited there the critics thereof and the disputes or controversies that have resulted. We explain that this, indeed interesting and relevant, issue would certainly need an individual paper with this particular aim. For our present paper, we preferred to delimit our study to the period 1980-2019 rather than dive on controversies about the reliability of data of earlier periods. 1. Alexandris, S. Review of Atmospheric Temperature and CO₂: Hen-or-Egg Causality? (Version 1). Sci 2020, https://www.mdpi.com/2413-4155/2/3/72 (posted and accessed on 26 September 2020). 2. Koutsoyiannis, D.; Kundzewicz, Z.W. Atmospheric Temperature and CO2: Hen-or-Egg Causality? (Version 1). Sci 2020, 2, 72. 3. Beck, E.-G. 180 Years of atmospheric CO2 gas analysis by chemical methods. Energy and Environment 2007, 18 (2), 259-282. Round 2 Reviewer 1 Report The manuscript is obviously improved after revision. I thank/appreciate the authors’ effort to explain every single detail on the theoretical basis and interpretation of the methodology and findings very clearly and thoroughly. I think the methodologies are robust. Overall, the authors’ have presented the whole story persuasively. The findings/conclusion of this work are somewhat contrasting to the public perception about the causes and effect of the global warming, I believe. The authors have explained their methodology/data and findings very clearly so that interested readers can easily reproduce the results and interpret them. Finally, this paper will open a new discussion on methodology/interpretation of the well-known problem and hence should be published. Author Response Once again, we thank Yog Aryal for his review, his positive evaluation and approval of Version 2 of our paper. We are very pleased that the Reviewer appreciated our effort, explanations and theoretical analyses, and that he found that we have presented the whole story persuasively and with robust methodology. We are particularly pleased by his comment that this paper would open a new discussion on methodology/interpretation. Indeed, we look forward to it. PS. We take the opportunity to thank the journal for the nice experience it offered us. We are particularly enthusiastic with the eponymous post-publication peer review system, which certainly is a most promising, revolutionary approach for the future of journal publishing. Reviewer 2 Report The authors have adopted most of my suggested revisions as well as those of the other reviewers. I think this revised version has significantly improved and is now a worthy addition to the scientific literature on this topic. With that in mind, I now recommend publication. However, I have a few additional comments which the authors might like to consider first. Specifically, while human activity may increase atmospheric CO2, the effect this might have on atmospheric temperatures (including surface air temperatures) depends on what the true value of the “climate sensitivity” is, i.e., the expected increase in global temperatures to a doubling of atmospheric CO2. As discussed in Sections 4 and 5 of the cited Connolly et al. (2020) paper, there is considerable ongoing debate over this. If the climate sensitivity is at the lower end of the values published in the literature, then it is plausible that the increases in CO2 over their study time period (1980-2019) might not have had a significant effect on T over that time frame. Rather, the changes in T over that period might have been determined by other factors, e.g., changes in solar activity. [With that in mind, see for instance our Soon, Connolly & Connolly, 2015 paper: Willie Soon, Ronan Connolly and Michael Connolly (2015). "Re-evaluating the role of solar variability on Northern Hemisphere temperature trends since the 19th century", Earth-Science Reviews, Volume 150, November 2015, Pages 409-452 https://doi.org/10.1016/j.earscirev.2015.08.010 ]. In that case, the “human activity → CO2 → T” causation would have been broken at the second stage, but the first stage could have still been valid, i.e., “human activity → CO2”. The analysis in their paper would still be relevant. Author Response Once again, we thank Ronan Connolly for his review, his positive evaluation and approval of Version 2 of our paper. We are very pleased that the Reviewer found Version 2 significantly improved and a worthy addition to the scientific literature on this topic. We appreciate his additional comments he offers us to consider, for which our responses follow. 1. We agree with the Reviewer that this is an issue to consider in our future research. Indeed, a single paper cannot study all folds of a subject that is as complex and also as important as the relationship of temperature and CO2 concentration. We are thus taking a note of the suggested paper for our future research. 2. We agree with the reviewer and we will follow the first suggestion he offers, i.e. to replace “is” with “might potentially be” in the statement in question. 3. Again, this is an issue that we will certainly consider in our future research, which we have already announced in Version 2 (see p. 24; “While we too are preparing a theoretical study, in which we will discuss in detail some theories …”. 4. We respectfully disagree with this comment for two reasons. First, the Appendices contain important information, which was put in Appendices to improve the readability of the body of the paper, not because it is supplementary information. Second, because some of the Appendices (in particular, Appendices A and D) have been created in response to review comments on the Version 1, and thus they belong to the reasons why the reviewer finds Version 2 “significantly improved”. Our choice to put this new information in Appendices does not at all mean that we devalue the reviewers’ comments. On the contrary, we believe it is evident that we have put a great effort to address these constructive comments. In other words, our choice of the paper structure is merely related to optimal readability of the paper. We are confident that other readers would have similar questions with the reviewers. Keeping the Appendices attached to the body of the paper these readers will conveniently find our answers. Reviewer 3 Report Dear editor, I had pointed out from in my first review that the submitted document is an original work with a high degree of documentation. Thus, I have nothing further to say. At the second stage, the authors have successfully answered all the questions/comments and they improved and enriched the document. I am very satisfied with the ending result of the manuscript and I think it will be interesting to the scientific community. I highly recommend the publication of the work in your journal. Author Response Once again, we thank Stavros Alexandris for his review, his positive evaluation and approval of Version 2 of our paper. We are very pleased that the Reviewer noted that we have successfully answered all the questions/comments and we improved and enriched our paper.
ReferencesReply to Reviewer 3 – Stavros Alexandris
References